# Optimizing Calibration by Gaining Aware of Prediction Correctness

## Abstract

Model calibration aims to align confidence with prediction correctness. The Cross-Entropy (CE) loss is widely used for calibrator training, which enforces the model to increase confidence on the ground truth class. However, we find the CE loss has intrinsic limitations. For example, for a narrow misclassification, a calibrator trained by the CE loss often produces high confidence on the wrongly predicted class (*e.g.*, a test sample is wrongly classified and its softmax score on the ground truth class is around 0.4), which is undesirable. In this paper, we propose a new post-hoc calibration objective derived from the aim of calibration. Intuitively, the proposed objective function asks that the calibrator decrease model confidence on wrongly predicted samples and increase confidence on correctly predicted samples. Because a sample itself has insufficient ability to indicate correctness, we use its transformed versions (*e.g.*, rotated, greyscaled, and color-jittered) during calibrator training. Trained on an in-distribution validation set and tested with isolated, individual test samples, our method achieves competitive calibration performance on both in-distribution and out-of-distribution test sets compared with the state of the art. Further, our analysis points out the difference between our method and commonly used objectives such as CE loss and Mean Square Error (MSE) loss, where the latters sometimes deviates from the calibration aim.

## 1 Introduction

Model calibration is an important technique to enhance the reliability of machine learning systems. Generally, it aims to align predictive uncertainty (a.k.a. confidence) with prediction accuracy. We are interested in post-hoc accuracy preserving calibrators that scale the model output to make it calibrated (Guo et al., 2017; Zhang et al., 2020; Tomani et al., 2022; Gupta et al., 2020; Kull et al., 2019).

Existing methods typically use Maximum Likelihood Estimation (MLE) to train a calibrator for the classification task, such as the Mean Square Error (MSE) loss (Tomani et al., 2022; 2023; Zhang et al., 2020) and the Cross-Entropy (CE) loss (Guo et al., 2017; Zou et al., 2023). Although these approaches demonstrate efficacy in reducing calibration errors such as Expected Calibration Error (ECE) and Brier scores, they lack theoretical guarantee that the calibration error is minimized when MLE converges. In Fig. 1, for an image which is incorrectly classified and has a relatively high probability on the ground truth class, calibrators trained by the CE or MSE loss would give high confidence on the wrongly predicted class. It means a user may trust this prediction to be true. As to be revealed in Sec. 3.4, the inherent problem of CE and MSE loss functions limits them in calibrating such test cases.

In this paper, we derive a concrete interpretation of the goal of calibration which is then directly translated into a novel loss function that aligns with the newly interpreted goal. Specifically, we start from the general definition of calibration and its error (Guo et al., 2017), and then, under a finite test set, represent the error in a discretized form. Minimizing this discretized error gives us a very interesting and intuitive calibration goal: A correct prediction should have possibly high confidence and a wrong prediction should have possibly low confidence. Theoretically, this optimization goal can naturally reduce the overlap of confidence values between correct and incorrect predictions.

We translate this goal into a loss function that enforces high confidence (*i.e.*, 1) for correctly classified samples, and low confidence (*i.e.*, $\frac{1}{C}$, where $C$ is the number of classes) for wrongly classified ones, named Correctness-Aware (CA) loss. Nevertheless, it is non-trivial to identify classification

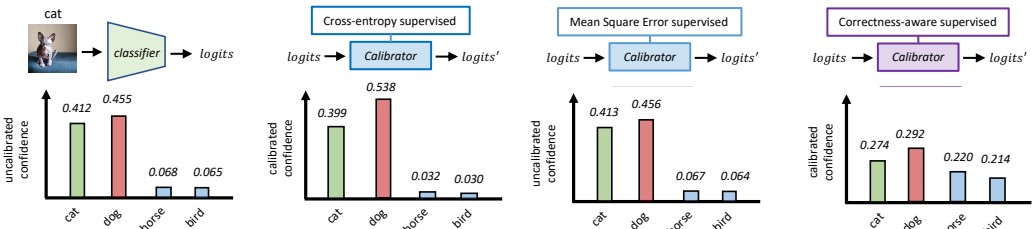

Figure 1: **A failure example for calibrators trained by the Cross-Entropy (CE) or Mean Square Error (MSE) loss.** A classifier makes a wrong prediction of a cat image. Before calibration, the classifier gives probabilities of 0.412 and 0.455 on the ground-truth and predicted classes, respectively. The calibrator trained with the CE loss assigns even higher confidence 0.538 to the wrong class, making things worse, and that trained by MSE maintains a similar confidence 0.456. In comparison, calibrator trained with the proposed Correctness-Aware (CA) loss effectively decreases confidence of this wrong prediction to 0.292, improving calibration.

correctness. We propose to use transformed versions of original images as the calibrator input: consistency between their prediction results suggests prediction correctness of the original sample.

Our method allows a calibrator to be trained on the in-distribution validation set and directly applied to individual test samples during inference. We demonstrate the effectiveness of the proposed strategy on various test sets. In both in-distribution (IND) and out-of-distribution (OOD) test sets, our method is clearly superior to uncalibrated models and competitive compared with state-of-the-art calibrators. Moreover, our method demonstrates the potential to better separate correct and incorrect test samples using their calibrated confidence. Below we summarize the main contributions.

- Theoretically, we derive the concrete goal of model calibration that has a clear semantic meaning. This allows us to design a new calibration loss function: correctly classified samples should have high confidence, while incorrectly classified ones with low confidence.
- To indicate prediction correctness, we use the softmax prediction scores of transformed versions of the original image as calibrator inputs.
- Our method achieves competitive calibration performance on various IND and OOD datasets.
- We diagnose commonly used calibration loss functions including the CE and MSE loss: they are often limited under test samples of certain characteristics.

## 2 RELATED WORK

**Loss functions used in post-hoc calibration.** Existing post-hoc calibrators typically use the Maximum Likelihood Estimation (MLE) for optimization (Jung et al., 2023; Tao et al., 2023). For example, Mukhoti et al. (2020) use the CE loss, while Kumar et al. (2019) use the MSE loss. Additionally, Mukhoti et al. (2020) uses the focal loss, a variant to CE, which enhances learning on wrong predictions. Guo et al. (2017) directly optimizes the Expected Calibration Error (ECE). This paper identifies inherent problems with MLE for calibration and derives a new loss function that better aligns with calibration goal.

**OOD calibration** deals with distribution shifts in test sets (Tomani et al., 2021). A useful practice is modifying the calibration set to cover OOD scenarios (Tomani et al., 2021; Krishnan & Tickoo, 2020), but these methods are usually designed for specific OOD scenarios which may lead to compromised IND calibration performance. Another approach adapts to test data distribution, whether OOD or IND, using domain adaptation (Wang et al., 2020), calibration sets from multiple domains (Gong et al., 2021), or improving calibration sets by estimating test set difficulty (Zou et al., 2023). In comparison, our calibrator is trained on the IND validation set only, and does not need test batches for update, but still demonstrates improved and competitive OOD calibration performance.

**Predicting classification correctness** has not been widely studied. Among the few, Xia & Bouganis (2023) investigate a three-way classification problem: classify a test sample into correct prediction,

Figure 2: **Calibration pipeline.** For a given image sample $\mathbf{X}$, we first obtain its logit vector $\mathbf{z}$ and softmax vector $\mathbf{v}$. We then apply $M$ different transformations (*e.g.*, rotation, greyscale, colorjitter, *etc.*) on $\mathbf{X}$ to get its transformed versions as well as their related softmax vectors as $\mathbf{v}_i$ ($i \in \mathcal{I}_M$). Indices $\mathbf{q} \in \mathbb{R}^k$ of the top-$k$ largest probabilities (softmax scores) in $\mathbf{v}$ are used to acquire top-$k$ scores from $\mathbf{v}_i$ to form the concatenated input $\oplus_{i \in \mathcal{I}_M} \mathbf{v}_i[\mathbf{q}]$ to the calibrator. The calibrator outputs a temperature $\tau$, then being used to update the logit vector $\mathbf{z}$ to produce the calibrated softmax vector. We use our proposed Correctness-Aware (CA) loss (Sec. 3.2).

wrong prediction, or an out-of-distribution sample (its category is outside the training label space). We find the task of predicting classification correctness closely connected to model calibration.

# 3 APPROACH

## 3.1 DEFINING CALIBRATION ERROR FROM ITS GOAL

**Notations.** We study calibration under the multi-way classification problem. Regular fonts are scalars, *e.g.*, $\tau$; vectors are denoted by lowercase boldface letters, *e.g.*, $\mathbf{x}$; matrices by the uppercase boldface, *e.g.*, $\mathbf{X}$ for an image.[1] $\mathcal{I}_C$ denotes an index set of integers $\{1, ..., C\}$, operator ';' and $\oplus$ concatenate vectors, *e.g.*, $\oplus_{i \in \mathcal{I}_M} \mathbf{v}_i = [\mathbf{v}_1; ...; \mathbf{v}_M]$. A classifier $f$ takes a $d$-dimensional input $\mathbf{x} \in \mathbb{R}^d$ and its corresponding label $y \in \mathcal{I}_C$ with $C$ classes which are sampled from the joint distribution $p(\mathbf{x}, y) = p(y|\mathbf{x})p(\mathbf{x})$. We use $\equiv$ to denote the equivalence. The output of $f$ is denoted as $f(\mathbf{X}) = (\hat{y}, \hat{c})$, where $\hat{y}$ and $\hat{c}$ denote the predicted class and maximum confidence score, respectively.

**Calibration goal.** According to Guo et al. (2017), the goal of model calibration is to "align confidence with the accuracy of samples." Based on this, existing literature define perfect calibration as:

$$\mathbb{P}(\hat{y} = y | \hat{c} = c) = c, \forall c \in [0, 1]. \tag{1}$$

**Our calibration error formulation.** We interpret Eq. (1) as: for any predicted confidence $\hat{c}$, the expected classification accuracy $\mathbb{E}_{\hat{c}}^{\text{acc}}$ of model $f$ on the conditional distribution $p(\mathbf{x}|\hat{c})$ should equal $\hat{c}$. Based on this interpretation, we write the calibration error of classifier $f$ as a function of $\hat{c}$:

$$l_f(\hat{c}) = D(\hat{c} \| \mathbb{E}_{\hat{c}}^{\text{acc}}) = \|\hat{c} - \mathbb{E}_{\hat{c}}^{\text{acc}}\| = \|\hat{c} - \int \mathbb{I}\{y_\mathbf{x} = \hat{y}_\mathbf{x}\} dp(\mathbf{x}|\hat{c})\|, \tag{2}$$

where $D$ denotes discrepancy measurement and $\| \cdot \|$ denotes a norm, such as the $\ell_2$- distance. The indicator function $\mathbb{I}\{\cdot\}$ returns 1 if the given condition (the prediction matches the ground truth label accurately) is true; otherwise, it returns 0.

Denoting the distribution of predicted confidence as $p(\hat{c})$ and its probability density function as $dp(\hat{c})$, the expectation of calibration error[2] of $f$ on $p(\hat{c})$ can be expressed as:

$$\mathbb{E}_f = \int l_f(\hat{c}) dp(\hat{c}), \tag{3}$$

where $l_f(\hat{c})$ is defined in Eq. (2). A model $f$ is considered to be perfectly calibrated if $\mathbb{E}_f = 0$. Model calibration is to optimize a calibrator which reduces $\mathbb{E}_f$ as much as possible.

## 3.2 CORRECTNESS-AWARE LOSS

In practice, the distribution $p(\hat{c})$ is unknown, hence, Eq. (3) cannot be directly computed. To solve this, we approximate the calibration error $\mathbb{E}_f$ by replacing $p(\hat{c})$ with an empirical distribution, formed

---

[1]For simplicity, we omit three color channels.

[2]This differs from the expected calibration error (ECE) metric. The ECE metric uses discretized histogram bins, whereas our calibration goal employs the continuous form.

by assembling Dirac delta functions (Dirac, 1981) centered at each predicted sample confidence $\hat{c}$ computed from a given dataset $\mathcal{D} = \{(\mathbf{x}_i, y_i)\}_{i=1}^n$, where $n$ is the number of samples:

$$dp(\hat{c}) = \frac{1}{n} \sum_{i=1}^n \delta_{\hat{c}_i}(\hat{c}). \tag{4}$$

Substituting Eq. (2) and Eq. (4) into Eq. (3), the empirical calibration loss can be written as:

$$\mathbb{E}_f^{\text{emp}} = \frac{1}{n} \sum_{i=1}^n \|\hat{c}_i - \int \mathbb{I}\{y_{\mathbf{x}_i} = \hat{y}_{\mathbf{x}_i}\} dp(\mathbf{x}_i | \hat{c}_i)\|. \tag{5}$$

However, Eq. (5) is still hard to compute because the probability density function $dp(\mathbf{x}_i | \hat{c}_i)$ is not accessible in practice. As such, we further discretize Eq. (5), where we assume a finite number of samples $\{\mathbf{x}_{ij}\}_{j=1}^m$ in each distribution $p(\mathbf{x}_i | \hat{c}_i)$. Consequently, $\mathbb{E}_f^{\text{emp}}$ is reformulated as:

$$\mathbb{E}_f^{\text{emp}} = \frac{1}{n} \sum_{i=1}^n \|\hat{c}_i - \frac{1}{m} \sum_{j=1}^m \mathbb{I}\{y_{\mathbf{x}_{ij}} = \hat{y}_{\mathbf{x}_{ij}}\}\|. \tag{6}$$

Note that in practice, $m = 1$, because there is only one test sample for each $p(\mathbf{x}_i | \hat{c}_i)$, *e.g.*, a dataset only has one test sample with confidence 0.52893.[3] Therefore, we define the correctness aware (CA) loss as the empirical calibration loss of classifier $f$ on a test set with $n$ samples:

$$\mathbb{E}_f^{\text{emp}} = \frac{1}{n} \sum_{i=1}^n \|\hat{c}_i - \mathbb{I}\{y_{\mathbf{x}_i} = \hat{y}_{\mathbf{x}_i}\}\|. \tag{7}$$

**Lower and upper bounds of CA loss.** Let us derive the CA loss range of Eq. (7) for a given test sample $\mathbf{x}_i$: (i) If the sample is correctly classified, the indicator function $\mathbb{I}^+\{y_{\mathbf{x}_i} = \hat{y}_{\mathbf{x}_i}\} = 1$. The maximum confidence range for a correctly classified sample is $\hat{c}_i^+ \in (\frac{1}{C}, 1]$, and the loss range would be $\|\hat{c}_i^+ - \mathbb{I}^+\{y_{\mathbf{x}_i} = \hat{y}_{\mathbf{x}_i}\}\| \in [0, 1 - \frac{1}{C})$. (ii) If the sample is wrongly classified, $\mathbb{I}^-\{y_{\mathbf{x}_i} = \hat{y}_{\mathbf{x}_i}\} = 0$ and $\hat{c}_i^- \in (\frac{1}{C}, 1]$, the loss range would be $(\frac{1}{C}, 1]$. Let $\rho \in [0, 1]$ be the ratio of correctly classified samples in $\mathcal{D}$. For the correctly classified samples, we have:

$$0 \leq \sum_{i=1}^{\rho \times n} \|\hat{c}_i^+ - \mathbb{I}^+\{y_{\mathbf{x}_i} = \hat{y}_{\mathbf{x}_i}\}\| < \rho \times n \times \left(1 - \frac{1}{C}\right), \tag{8}$$

and for the wrongly classified samples, we have:

$$(1 - \rho) \times n \times \frac{1}{C} < \sum_{i=1}^{(1-\rho) \times n} \|\hat{c}_i^- - \mathbb{I}^-\{y_{\mathbf{x}_i} = \hat{y}_{\mathbf{x}_i}\}\| \leq (1 - \rho) \times n. \tag{9}$$

Combining Eq. (8) and (9) with Equation (7), we can derive the lower and upper bounds of CA loss:

$$\mathbb{E}_f^{\text{emp}} = \frac{1}{n} \sum_{i=1}^n \|\hat{c}_i - \mathbb{I}\{y_{\mathbf{x}_i} = \hat{y}_{\mathbf{x}_i}\}\|,$$

$$n \times \mathbb{E}_f^{\text{emp}} = \sum_{i=1}^n \|\hat{c}_i - \mathbb{I}\{y_{\mathbf{x}_i} = \hat{y}_{\mathbf{x}_i}\}\| = \sum_{j=1}^{\rho \times n} \|\hat{c}_j^+ - \mathbb{I}^+\{y_{\mathbf{x}_j} = \hat{y}_{\mathbf{x}_j}\}\| + \sum_{k=1}^{(1-\rho) \times n} \|\hat{c}_k^- - \mathbb{I}^-\{y_{\mathbf{x}_k} = \hat{y}_{\mathbf{x}_k}\}\|, \tag{10}$$

$$n \times \left(\frac{1-\rho}{C}\right) \leq n \times \mathbb{E}_f^{\text{emp}} \leq n \times \rho \times \left(1 - \frac{1}{C}\right) + n \times (1 - \rho) = n \times \left(\frac{1-\rho}{C} + \frac{C-1}{C}\right),$$

$$\frac{1-\rho}{C} \leq \mathbb{E}_f^{\text{emp}} \leq \frac{1-\rho}{C} + \frac{C-1}{C}, \tag{11}$$

where $C$ is the number of classes. Hence, the CA loss has a lower bound of $\frac{1-\rho}{C}$ and an upper bound of $\frac{1-\rho}{C} + \frac{C-1}{C}$. We observe that both lower and upper bounds are closely tied to $\frac{1-\rho}{C}$, representing the fraction of misclassified samples per class in whole $\mathcal{D}$.

---

[3]We have a mild assumption that a test set has few, if any, duplicated images, where Eq. (6) will also approximately hold.

**Theoretical insights of CA loss.** First, we rewrite Eq. (7) as a sum of two CA loss components, $\mathbb{I}\{y_{\mathbf{x}_i} = \hat{y}_{\mathbf{x}_i}\}$ equals 1 and 0 for correctly and incorrectly classified samples, respectively, and we use $\ell_1$ as discrepancy measurement for simplicity:

$$\mathbb{E}_f^{\text{emp}} = \frac{1}{n}\left(\sum_{j=1}^{\rho \times n}(1 - \hat{c}_j^+) + \sum_{k=1}^{(1-\rho) \times n}\hat{c}_k^-\right), \tag{12}$$

where $\hat{c}_j^+$ and $\hat{c}_k^-$ denote, respectively, the maximum confidence scores for correctly and incorrectly classified samples. Suppose $\rho \geq 50\%$, we have:

$$\mathbb{E}_f^{\text{emp}} = \frac{1}{n}\left(\sum_{j=1}^{(1-\rho) \times n}(1 - \hat{c}_j^+) + \sum_{j=(1-\rho)n+1}^{\rho \times n}(1 - \hat{c}_j^+) + \sum_{k=1}^{(1-\rho) \times n}\hat{c}_k^-\right)$$

$$= 1 - \rho + \frac{1}{n}\left(\sum_{k=1}^{(1-\rho) \times n}\hat{c}_k^- - \sum_{j=1}^{(1-\rho) \times n}\hat{c}_j^+\right) + \frac{1}{n}\sum_{j=(1-\rho)n+1}^{\rho \times n}(1 - \hat{c}_j^+). \tag{13}$$

Now using $\mathbb{E}^{\text{diff}} = \frac{1}{(1-\rho) \times n}\left(\sum_{k=1}^{(1-\rho) \times n}\hat{c}_k^- - \sum_{j=1}^{(1-\rho) \times n}\hat{c}_j^+\right)$ to denote the expectation of the difference in maximum confidence values between correct and incorrect predictions, and $\mathbb{E}^+ = \frac{1}{(2\rho-1)n}\sum_{j=(1-\rho)n+1}^{\rho \times n}(1 - \hat{c}_j^+)$ to denote the expectation of the maximum softmax scores for the rest $(2\rho - 1)n$ correctly classified samples, Eq. (13) can be written as:

$$\mathbb{E}_f^{\text{emp}} = (1-\rho)\mathbb{E}^{\text{diff}} + (2\rho-1)\mathbb{E}^+ + (1-\rho) \equiv (1-\rho)\mathbb{E}^{\text{diff}} + (2\rho-1)\mathbb{E}^+. \tag{14}$$

We notice that minimizing our CA loss is equivalent to minimizing either $\mathbb{E}^{\text{diff}}$ or $\mathbb{E}^+$ (omitting the constant $(1-\rho)$): (i) minimizing $\mathbb{E}^{\text{diff}}$ aims to maximize the expectation of the difference in maximum confidence scores between correct and incorrect predictions, and (ii) minimizing $\mathbb{E}^+$ aims to push the maximum confidence score of correctly classified samples to 1. These two objectives align well with the model calibration goal (Sec. 3.1). Below we take a close look at $\mathbb{E}^{\text{diff}}$:

$$\mathbb{E}^{\text{diff}} = \frac{1}{(1-\rho) \times n}\left(\sum_{k=1}^{(1-\rho) \times n}\hat{c}_k^- - \sum_{j=1}^{(1-\rho) \times n}\hat{c}_j^+\right) = \frac{1}{1-\rho}\left(\mathbb{E}_f^{\text{emp}} - \frac{1}{n}\sum_{j=(1-\rho)n+1}^{\rho \times n}(1 - \hat{c}_j^+) + \rho - 1\right),$$

$$\mathbb{E}_f^{\text{emp}} \to \frac{1-\rho}{C} \equiv \mathbb{E}^{\text{diff}} \to \frac{\left(\frac{1}{C} - 1\right)\rho}{1-\rho} < 0. \tag{15}$$

Eq. (15) demonstrates that minimizing our CA loss $\mathbb{E}_f^{\text{emp}}$ during training toward the lower bound $\frac{1-\rho}{C}$ is equivalent to pushing $\mathbb{E}^{\text{diff}}$ toward $\frac{\left(\frac{1}{C} - 1\right)\rho}{1-\rho} < 0$. This means pushing the average maximum softmax scores of wrongly classified samples away from those of correctly classified samples, thereby reducing the overlap of confidence values between correct and incorrect predictions.

## 3.3 GAINING CORRECTNESS AWARENESS

From the intuition of the CA loss in Eq. (7), its optimization requires the post-hoc calibrator to be aware of the correctness of each test sample. Empirically, we find the test sample itself offer limited help to distinguish correctness, which leads to undesirable calibration performance.

Our approach, inspired by (Deng et al., 2022), leverages the discovery that consistency in model predictions for transformed images correlates strongly with accuracy. While their insight focuses on dataset-level consistency, our assumption extends to individual samples: the model's behavior on transformed samples informs prediction correctness.

The pipeline of our calibrator is presented in Fig. 2, we aim to calibrate a classification model $f$. To do so, we compute the logit vector $\mathbf{z}$, softmax vectors of an original image $\mathbf{X}$ and its transformed versions $\mathbf{v}$ and $\mathbf{v}_i$ ($i \in \mathcal{I}_M$, assuming $M$ types of transforms), respectively. We determine the indices $\mathbf{q} \in \mathbb{R}^k$ of the $k$ largest softmax scores of $\mathbf{v}$, and use these indices $\mathbf{q}$ to locate and select the corresponding values from $\mathbf{v}_i$, forming new vector with $k$ dimensions. These $k$-dim vectors $\mathbf{v}_i[\mathbf{q}]$ of the transformed images are concatenated as $\oplus_{i \in \mathcal{I}_M}\mathbf{v}_i[\mathbf{q}]$, and used as the calibrator input.

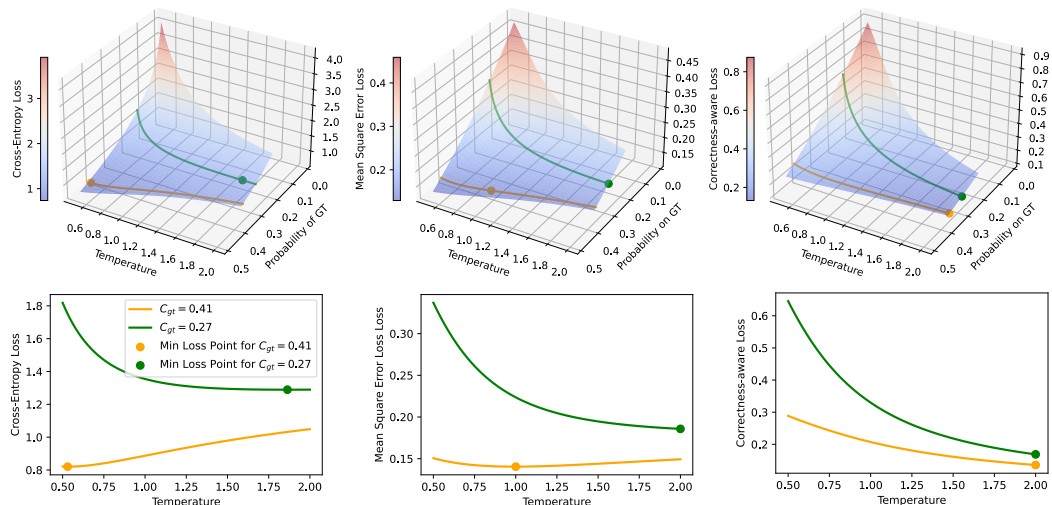

Figure 3: **Comparison of different loss functions w.r.t. temperature and softmax probability of the ground truth (GT) class.** In a four-way classification task, we examine a wrongly predicted sample with logit vector $[a, 2.0, 0.1, 0.05]$, where $a < 2$ is the value on the ground truth class. We use $c_{\text{gt}}$ to donate the softmax score of the GT class. **Top:** The loss surface plots for varying temperatures and $c_{\text{gt}}$, with red and blue arrows representing positive and negative temperature gradients, respectively. **Bottom:** Shows 2D loss curves for varying $c_{\text{gt}}$. The lines in the bottom charts correspond to the lines of the same color in the top charts. Compared with Maximum Likelihood Estimation (MLE) based functions (*e.g.*, Cross-Entropy, Mean Squared Error), our Correctness-Aware loss minimization does not favor temperatures below 1 for incorrect predictions, while sometimes MLE does.

Calibrator, $g$ parameterized by $\theta$, which is trained by the proposed CA loss. Here, the calibrator consists of two fully connected layers with a ReLU activation function in between. Each hidden layer comprises 5 nodes. The calibrator is optimized on the calibration (a.k.a., validation) set, and the calibration output is temperature $\tau$ to be used to scale the model logit vector $\mathbf{z}$ of original image. Below we show these steps in equations:

$$\mathbf{q} = \operatorname*{argmax}_{\mathbf{q}} \mathbf{v} \in \mathbb{R}^k, \tag{16}$$

$$\tau = g_\theta(\oplus_{i \in \mathcal{I}_M} \mathbf{v}_i[\mathbf{q}]), \tag{17}$$

$$\hat{c} = \max_c \sigma(\mathbf{z}/\tau)^{(c)}, \tag{18}$$

where $\sigma(\cdot)$ denotes the softmax function. Eq. (18) retrieves the maximum softmax prediction score $\hat{c}$. Based on the CA loss in Eq. (7), the optimization goal now becomes:

$$\arg\min_\theta \mathbb{E}_f^{\text{emp}}(\hat{c}, y_{\mathbf{x}}, \hat{y}_{\mathbf{x}}), \tag{19}$$

where $y_{\mathbf{x}}$ and $\hat{y}_{\mathbf{x}}$ denote respectively the ground truth and predicted labels.

In practice, transformations can be grayscale, rotation, color jitter, adding Gaussian noise, random erasing, *etc*. They are applied to the original image during training and inference. We do not assume access to test batches, which are used in some previous works (Guo et al., 2017; Wang et al., 2023). Nevertheless, if we assume such access, we can retrieve from test batches images that are similar to the original one and use softmax vectors of the retrieved images as calibrator inputs. As to be shown in Sec. 5, we find that grayscale, rotation, and colorjitter are effective ones, and that using four transformations give a good trade-off between calibration performance and computational cost.

During inference, using the $k$-dim vectors from the transformed images, we obtain an adjusted temperature from the calibrator. This temperature is used to scale the logits of the original image, the softmax vector of which is then updated. Alg. 1 in Appendix A summarises this calibration process.

### 3.4 COMPARISON BETWEEN CA LOSS AND MLE

Maximum Likelihood Estimate (MLE) is widely used for calibration training (Kumar et al., 2019), under concrete formats such as the Cross-Entropy (CE) or Mean Square Error (MSE) losses. This section goes through their connections and differences with the CA loss. Appendix B shows more discussions.

**For correct predictions, MLE (*e.g.*, CE or MSE loss) has similar effect with the CA loss**. MLE enforces the softmax probability of the ground-truth class to be close to 1. For correct predictions, the softmax probability of the ground-truth class equals the sample confidence (maximum probability in the softmax vector). Under this scenario, MLE aligns with both the calibration objective and the CA loss: the confidence of correct predictions should be possibly high.

**For wrong predictions, MLE sometimes deviates from the calibration goal while CA is theoretically consistent.** In Fig. 3, we visualize the loss surface of the CE, MSE, and CA loss w.r.t. temperature and softmax probability on the ground truth class (first row), from which we use examples of two typical calibration training samples for more intuitive illustration (both first and second rows). Particularly, optimal temperature ($x$-axis) is achieved when the respective loss ($y$ axis) is minimum (the second row is easier to read).

For *an absolutely wrong sample* (green curves in both rows), whose probability of the wrongly predicted class is far greater than that on the ground-truth class, the optimization direction of MLE is similar to CA: the loss curve keeps decreasing and finally a large temperature or a low confidence is obtained. In fact, under this scenario, calibration objective requires the probability of the ground-truth class to increase and probability of the wrongly predicted class to decrease. This is consistent with the objective of MLE: to increase probability on the ground-truth class.

For *a narrowly wrong sample* (yellow curves in both rows), whose probability on the wrongly predicted class is much closer to that on the ground-truth class, the optimization direction of MLE is very different from or even opposite to CA. Take the yellow curves in the second row of Fig. 3 as example. The CE loss, to become smaller, leads to a small temperature, meaning a large confidence, which is undesirable for this wrongly predicted sample. For MSE, its minimum is achieved when temperature is around 1.0, which does not change the temperature and confidence much. This again is undesirable. In comparison, the CA loss keep decreasing when temperature increases so will eventually give a large temperature or a small confidence for this type of samples. This is consistent with the calibration objective.

Empirically, we find that such narrowly wrong predictions take up 2%-8% of the calibration set (ImageNet validation).[4] This would negatively impact training efficacy of MLE. Moreover, during inference, if a test set has many such narrowly wrong predictions, MLE will also be negatively impacted because of its unsuitable in dealing with such samples during training. This would explain why our system is superior to and on par with state of the art on OOD and IND test sets, respectively (refer to Sec. 4). In both IND and OOD scenarios, our calibrated models are much better.

## 4 EXPERIMENTS

### 4.1 MODELS AND DATASETS

**ImageNet-1k setup.** 1. Models. We use 10 models trained or fine-tuned on the ImageNet-1k training set (Deng et al., 2009). We source these models from the model zoo Timm (Wightman, 2019). 2. Calibration sets. We use ImageNet-Val (Deng et al., 2009) to train calibrator. 3. Test sets. (1) *ImageNet-A(dversarial)* (Hendrycks et al., 2021b) comprises natural adversarial examples that are unmodified and occur in the real world. (2) *ImageNet-S(ketch)* (Wang et al., 2019) contains images with a sketch-like style. (3) *ImageNet-R(endition)* (Hendrycks et al., 2021a) comprises of 30,000 images that exhibit diverse styles. (4) *ObjectNet* (Barbu et al., 2019) is a real-world test set for object recognition where illumination, backgrounds and imaging viewpoints are very challenging. (5) *ImageNet-Val*. We train the calibrator on half of the ImageNet validation set and test it on the remaining half.

---

[4]We first compute the ratio of the probability on the ground-truth class to that on the wrongly predicted class. We define a sample is narrowly wrong prediction if this ratio is higher than 0.5.

Table 1: **Calibrator comparison under the ImageNet setup.** Each reported number is averaged over 10 classifiers, described in Sec. 4.1. We use five test sets: ImageNet-Val, ImageNet-A, ImageNet-R, ImageNet-S, and ObjectNet, and four metrics: ECE (bin=25), BS, KS and AUC (AUROC). Best results in each column are in bold. When comparing CA and CE, better results are in blue color.

| Method | ImageNet-Val | | | | ImageNet-A | | | | ImageNet-R | | | | ImageNet-S | | | | ObjectNet | | | |
|---|---|---|---|---|---|---|---|---|---|---|---|---|---|---|---|---|---|---|---|---|
| | ECE↓ | BS↓ | KS↓ | AUC↑ | ECE↓ | BS↓ | KS↓ | AUC↑ | ECE↓ | BS↓ | KS↓ | AUC↑ | ECE↓ | BS↓ | KS↓ | AUC↑ | ECE↓ | BS↓ | KS↓ | AUC↑ |
| Uncal | 8.71 | 14.01 | 14.39 | 85.96 | 39.44 | 32.90 | 43.47 | 61.87 | 13.97 | 16.89 | 19.90 | 88.06 | 20.92 | 21.67 | 29.01 | 82.57 | 31.21 | 25.20 | 36.48 | 78.05 |
| TS | 9.10 | 13.85 | 13.26 | 85.41 | 29.24 | 23.24 | 32.50 | 62.86 | 6.28 | 13.80 | 14.51 | 88.27 | 8.92 | 16.11 | 19.18 | 83.22 | 19.70 | 17.97 | 26.92 | 78.35 |
| ETS | 3.22 | 12.61 | 12.43 | 85.90 | 32.40 | 26.21 | 36.98 | 62.68 | 8.29 | 14.21 | 17.09 | 88.22 | 14.79 | 17.47 | 23.72 | 83.17 | 25.13 | 20.61 | 31.11 | 78.28 |
| MIR | 2.36 | 12.51 | 12.93 | 85.92 | 34.70 | 26.92 | 39.37 | 61.87 | 10.33 | 14.71 | 19.43 | 88.01 | 17.39 | 18.28 | 26.41 | 82.55 | 27.56 | 21.29 | 33.48 | 78.02 |
| SPL | 2.38 | 12.50 | 12.75 | 85.94 | 33.61 | 26.71 | 38.28 | 61.87 | 9.41 | 14.58 | 18.33 | 88.02 | 16.48 | 18.09 | 25.45 | 82.56 | 26.44 | 21.09 | 32.27 | 78.37 |
| Adaptive TS | 6.35 | 14.20 | 11.04 | 82.94 | 29.30 | 23.61 | 32.99 | 61.43 | 5.65 | 14.29 | 14.68 | 86.99 | 9.65 | 16.96 | 20.01 | 81.01 | 20.77 | 18.99 | 27.79 | 76.95 |
| TCP | 8.30 | 17.38 | 17.45 | 72.15 | 28.07 | 19.62 | 32.05 | 47.57 | 8.81 | 22.59 | 21.16 | 62.77 | 9.79 | 21.69 | 25.56 | 56.95 | 21.79 | 18.58 | 31.25 | 72.57 |
| ProCal | 2.89 | 12.60 | 13.33 | 86.08 | 38.82 | 30.99 | 42.35 | 61.75 | 11.84 | 16.29 | 19.13 | 86.22 | 19.85 | 19.63 | 28.10 | 82.30 | 25.22 | 22.61 | 31.58 | 75.15 |
| CE only (PTS) | 5.02 | 12.50 | 11.40 | 86.69 | 41.23 | 32.02 | 45.60 | 60.85 | 19.14 | 18.70 | 26.44 | 87.05 | 14.39 | 17.01 | 24.82 | 82.75 | 32.89 | 24.85 | 38.18 | 77.77 |
| CA only | 2.22 | 12.25 | 12.63 | 86.74 | 32.14 | 25.28 | 36.47 | 61.08 | 11.62 | 15.79 | 20.44 | 86.86 | 5.49 | 15.29 | 14.97 | 82.50 | 22.09 | 18.59 | 28.99 | 77.89 |
| CE+trans. | **3.42** | 12.87 | 11.77 | 85.36 | 28.06 | 22.01 | 32.38 | 63.47 | 6.22 | 13.28 | 15.30 | 88.64 | 11.52 | 15.94 | 21.08 | 83.84 | 20.89 | 18.28 | 27.69 | 78.50 |
| CA+trans. (ours) | 4.63 | **11.85** | **11.55** | **87.44** | **20.65** | **16.79** | **22.50** | **63.74** | **4.91** | **12.21** | **10.12** | **90.22** | **4.00** | **13.83** | **13.12** | **84.87** | **10.33** | **14.59** | **18.72** | **79.25** |
| CA + CE + trans | 4.24 | 13.15 | 11.53 | 84.91 | 26.54 | 20.85 | 30.80 | 63.85 | 5.63 | 12.86 | 14.57 | 88.96 | 9.50 | 15.15 | 19.52 | 84.31 | 21.42 | 18.79 | 28.02 | 78.14 |

Table 2: **Calibrator comparison under the CIFAR-10 setup.** Each number is averaged over 10 classifiers (see Sec. 4.1). We use one IND test set (CIFAR10.1) and three OOD test sets (CIFAR-10.1, CINIC, and CIFAR-10-C). Other notations are the same as Table 1.

| Method | CIFAR-10.1 | | | | Gaussian Blur | | | | Defocus Blur | | | | CINIC | | | |
|---|---|---|---|---|---|---|---|---|---|---|---|---|---|---|---|---|
| | ECE↓ | BS↓ | KS↓ | AUC↑ | ECE↓ | BS↓ | KS↓ | AUC↑ | ECE↓ | BS↓ | KS↓ | AUC↑ | ECE↓ | BS↓ | KS↓ | AUC↑ |
| Uncal | 10.22 | 12.59 | 13.80 | 85.08 | 45.72 | 43.48 | 50.01 | 66.7 | 34.79 | 34.79 | 39.73 | 71.49 | 24.25 | 25.29 | 28.42 | 78.76 |
| TS | 4.84 | 11.06 | 11.78 | 85.15 | 35.25 | 34.54 | 42.81 | 66.78 | 24.99 | 28.17 | 33.93 | 71.40 | 16.04 | 20.77 | 24.17 | **79.15** |
| ETS | 2.77 | **10.78** | 10.87 | 85.08 | 30.04 | 30.84 | 39.41 | 66.69 | 19.98 | 25.69 | 39.73 | 71.23 | 11.48 | 19.13 | 22.15 | 79.25 |
| MIR | **2.29** | 10.83 | 11.01 | 84.98 | 30.39 | 30.83 | 39.73 | 66.67 | 20.31 | 25.63 | 31.51 | 71.41 | 12.58 | 19.59 | 22.86 | 78.68 |
| SPL | 3.04 | 10.89 | **10.61** | 85.04 | 29.12 | 30.57 | 38.49 | 66.72 | 19.64 | 25.53 | 30.46 | 71.48 | 11.88 | 19.51 | 22.03 | 78.72 |
| Adaptive TS | 4.17 | 11.29 | 11.74 | 83.40 | 22.07 | 26.58 | 34.44 | 65.97 | 12.15 | 23.31 | 27.27 | 70.27 | 10.73 | 19.48 | 21.99 | 77.76 |
| TCP | 11.40 | 13.83 | 12.78 | 76.09 | 12.38 | 24.64 | 30.48 | 54.83 | 6.71 | 24.66 | 24.35 | 58.25 | 15.35 | 23.62 | 17.47 | 73.16 |
| ProCal | 2.96 | 11.27 | 11.60 | 82.85 | 37.26 | 36.83 | 44.28 | 64.50 | 25.68 | 28.99 | 34.36 | 69.39 | 18.04 | 22.19 | 25.14 | 76.22 |
| CE Only (PTS) | 2.92 | 10.85 | 11.26 | 85.12 | 31.01 | 31.24 | 40.07 | 67.28 | 21.01 | 25.86 | 31.80 | 71.95 | 13.69 | 19.92 | 23.27 | 78.93 |
| CA Only | 2.42 | 10.87 | 11.06 | 84.82 | 29.41 | 30.19 | 39.05 | 67.11 | 19.45 | 25.17 | 30.99 | **72.04** | 12.65 | 19.68 | 22.93 | 78.55 |
| CE+trans. | 2.87 | 10.90 | 11.22 | 84.84 | 18.49 | 24.51 | 32.22 | 67.18 | 9.09 | 22.04 | 25.79 | 71.52 | **7.05** | **18.31** | 20.65 | 79.03 |
| CA+trans. (ours) | 2.76 | 10.79 | 10.72 | **85.15** | **11.45** | **22.18** | **27.91** | 67.33 | **4.65** | **21.30** | **22.42** | 71.57 | 7.27 | 18.34 | **15.90** | 78.91 |

**CIFAR-10 setup.** 1. Models. We use 10 different models trained on the training split of CIFAR-10 (Krizhevsky et al., 2009) in this setup. We follow the practice in (Deng et al., 2022) to access the model weights. 2. Calibration set. Calibrators are trained on the test set of CIFAR-10. 3. Test sets. (1) *CINIC-10* (Darlow et al., 2018) is a fusion of both CIFAR-10 and ImageNet-C (Hendrycks & Dietterich, 2019) image classification datasets. It contains the same 10 classes as CIFAR-10. (2) *CIFAR-10-C(orruptions)* contains subsets from CIFAR-10 modified by perturbations such as blur, pixelation, and compression artifacts at various severities.

**iWildCam setup.** 1. Model. We use 10 models trained on the iWildCam(Beery et al., 2020) training set. They are downloaded from the official dataset website. 2. Calibration set. We train the calibrator on the iWildCam validation set. 3. Test set. We use the iWildCam test set containing animal pictures captured in the wild. Further details of the three setups are provided in Appendix C.

## 4.2 CALIBRATION METHODS AND EVALUATION METRICS

**Methods.** We compare our method with six popular calibration methods. They include scaling-based methods such as temperature scaling (TS) (Guo et al., 2017), ensemble temperature scaling (ETS) (Zhang et al., 2020), adaptive temperature scaling (Adaptive TS) (Joy et al., 2023), and parameterized temperature scaling (PTS) (Tomani et al., 2022). We also compare with binning method multi-isotonic regression (MIR) (Zhang et al., 2020), True Class Probability (TCP) (Corbiere et al., 2021), spline-based re-calibration method (Spline) (Gupta et al., 2020), and Proximity-Informed Calibration (ProCal) (Xiong et al., 2024).

**Metrics.** Apart from the expected calibration error (ECE) (Guo et al., 2017), we report the Brier score (BS), adaptive calibration error (ACE) (Nixon et al., 2019), and Kolmogorov-Smirnov (KS) error (Gupta et al., 2020). In addition, we use area under the ROC curve (AUROC) to evaluate how well the calibrators separate correct predictions from wrong predictions, which might also be a good metric for calibration (Appendix B). All the numbers in Table 1, 2, and 3 are averaged over results of calibrating 10 different models.

### 4.3 MAIN OBSERVATIONS

**Comparison of calibration performance with the state of the art.** We summarize calibration results under the ImageNet, CIFAR-10, and iWildCam setups in Table 1, 2, and 3, respectively. We have two observations. **First**, on OOD test sets, our method is very competitive across various metrics. For example, when compared with the second-best method on Object-Net, the ECE, BS, and KS metrics of our method are 10.56%, 3.69%, and 8.97% lower, respectively. **Second**, on near IND or IND test sets such as ImageNet-Val and CIFAR-10.1 , our method is less advantageous but is still competitive. The reason for our method being more effective on OOD test sets is that there are more narrowly wrong predictions, mentioned in the last paragraph in Sec. 3.4. Besides, as explained in Sec. 3.3, the use of transformed images might not be an optimal way to inform classification correctness.

Table 3: **Calibrator comparison under iWildCam setup.** Each number is averaged over 10 classifiers (see Sec. 4.1). We use the iWildCam test set. Other notations are the same as Table 1.

| Method | ECE↓ | BS↓ | KS↓ | AUC↑ |
|---|---|---|---|---|
| Uncal | 16.04 | 17.77 | 22.11 | 86.59 |
| TS | 6.63 | 14.06 | 14.90 | 86.22 |
| ETS | 6.83 | 14.05 | 13.84 | 86.17 |
| MIR | **5.01** | 13.53 | 12.57 | 86.61 |
| SPL | 5.98 | 13.70 | 12.44 | 86.63 |
| ProCal | 7.09 | 14.68 | 16.39 | 84.70 |
| Adaptive TS | 7.17 | 14.25 | 12.88 | 86.33 |
| TCP | 13.30 | 13.57 | 13.73 | 90.51 |
| CE only (PTS) | 8.73 | 17.90 | 17.45 | 78.65 |
| CA only | 8.07 | 17.54 | 16.95 | 79.15 |
| CE+trans. | 6.78 | 12.88 | 12.07 | 88.70 |
| CA+trans. (ours) | 7.21 | **11.81** | **10.24** | **90.52** |

**Comparing CA with MLE.** In Table 1, 2, and 3, we compare 'CA only' with 'CE only', and 'CA+trans' with 'CE+trans'. First, 'CA only' consistently outperforms 'MSE only' in 18 out of 20 scenarios under the ImageNet setup, 13 out of 16 scenarios under the CIFAR-10 setup, and 4 out of 4 scenarios under the iWildCam setup. Second, in most cases (*e.g.*, 19 out of 20 scenarios under ImageNet setup), 'CA+trans' is better than 'CE+trans'. In addition, in Table 1, we observe that the combination of CE and CA does not yield better results compared to using CA alone in the OOD test set. The superiority of the CA loss is more evident on OOD datasets as discussed in Appendix B.

**Potential of CA in allowing confidence to better separate correct and wrong predictions.** In Tables 1, 2, and 3, we compare separability and have two observations. First, existing methods typically do not have improvement in AUROC. This is not surprising, because their working mechanisms are not relevant to the separation of correct and wrong predictions. Second, our method improves AUROC under the ImageNet and iWildCam setups and in on par with existing methods on CIFAR-10. In fact, we find predictions of transformed images offer much less diversity under CIFAR-10 classifiers, losing their efficacy in telling prediction correctness. This could be addressed with a better method than transformed images, and we leave it for future work. These results, especially those under the challenging ImageNet and iWildCam setups, suggest our method has the potential to better distinguish between correct and wrong predictions by confidence scores, which could lead to improved decision making. A closer look at the ROC curves is provided in Fig. 4 and Appendix D.

**Effectiveness of using transformed images as calibrator input.** We compare 'CA+trans' and 'CA only' in Table 1, 2, and 3. It is very clearly that 'CA+trans' gives consistently better calibration performance than 'CA only'. It indicates the necessity of using transformed images.

## 5 FURTHER ANALYSIS

**Impact of narrowly wrong predictions in training and testing.** We construct various calibration (training) sets and test sets with samples of controlled degrees of being wrongly predicted. From Fig. 5 (left), if a test set is dominantly filled with narrowly wrong predictions, our method will have a huge improvement over CE and no calibration: in fact, CE has the same performance as no calibration in this scenario. As more absolutely wrong samples are included, the gap between smaller,

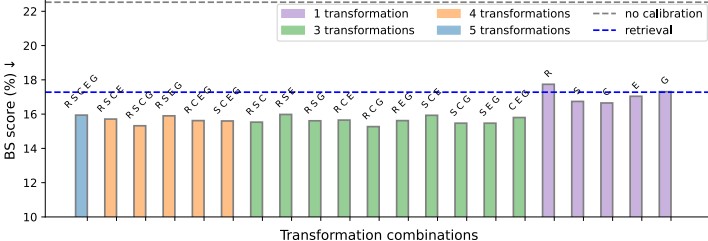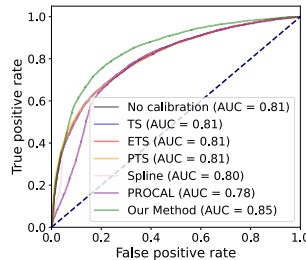

Figure 4: **(Left:)** Comparing various combinations of image transformations, including rotation (R), grayscale (S), colorjitter (C), random erasing (E) and Gaussian noise (G). Different colors means different numbers of transformations. Dashed lines denote performance of no calibration and retrieval-based augmentation that accesses test batches. **(Right:)** Visualization of ROC curves of various calibrators. Existing methods typically do not improve AUC, while our method effectively does. All results in this figure are reported for ObjectNet using the model 'beit_base_patch16_384', as introduced in Appendix C.

but our method is still superior. This is because the calibration set also has various degrees of wrong predictions, so CE is not as well trained as CA and actually has similar performance as no calibration.

On the other hand, from Fig. 5 (right), when training set contains lots of narrowly wrong predictions, CE is very poor and even worse then no calibration. When more absolutely wrong samples are included, CE becomes gradually better and even close to our method. These results empirically verifies our discussion in Sec. 3.4.

**Comparing different image transformations**. We try different combinations of image transformations (including retrieval-based augmentation in test batches) as calibrator input. Results are summarized in Fig. 4. We observe that using rotation, gray-scaling, and color-jittering generally give good calibration results. Retrieval-based augmentation is also competitive, but it requires access to test batches which might not be practical. Moreover, we find that using only one transformation is not ideal. While using more transformations is effective, three is a good number to balance between calibration performance and computational cost.

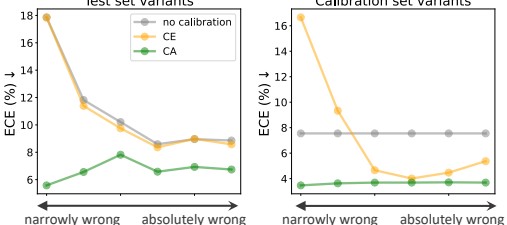

Figure 5: **Impact of narrowly wrong and absolutely wrong predictions on calibrator performance.** (**Left:**) we craft test sets containing 500 wrongly predicted samples with various degrees of being wrong. For example, the leftmost test set contains narrowly wrong samples, while the rightmost one contains absolutely wrong sample. Calibrator is trained on ImageNet-Val. (**Right:**) we craft training sets containing 1,000 wrong predictions and 1,000 correct predictions. The wrongly predicted samples also have different degrees of being wrong. We use ImageNet-A as test set. For both subfigures, we use *'beit_base'* as the classifier and compare CA with CE and no calibration. Our method is more superior when training/test sets contain more narrowly wrong predictions.

## 6 CONCLUSION

This paper starts from the general goal of calibration, mathematically interprets it, and derives a concrete loss function for calibration. Name as correctness-aware (CA) loss, in training it requires correct (wrong) predictions to have high (low) confidence, where such correctness is informed by transformed versions of original images. During inference, our calibrator also takes transformed images as input and tends to give high (low) confidence to likely correctly (wrongly) predicted images. We show our method is very competitive compared with the state of the art and potentially benefits decision making with plausible results on better separability of correct and wrong predictions. Moreover, we reveal the limitations of the CE and MSE losses for certain type of samples in the calibration set. Rich insights are given *w.r.t* how our method deals with such samples. In future we will study more effective correctness prediction methods to improve our system and how our method can be used for training large vision language models.

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

## A  OUR ALGORITHM

Alg. 1 shows the calibrator with our proposed Correctness-Aware loss.

---

**Algorithm 1** Calibrator with our proposed Correctness-Aware Loss

---

**Input**: a classification model $f$ to be calibrated ($f'(\cdot)$ extracts the logit vector, and $\sigma(\cdot)$ denotes the softmax function), a calibrator $g$ parameterized by $\theta$, the total number of transforms $M$, and $k$ for selecting the top-$k$ maximum softmax scores. $y$ and $\hat{y}$ denote the ground truth label and the predicted label for a given image sample $\mathbf{X}$, respectively.

**Step 1:** Obtain the logit vector: $\mathbf{z} = f'(\mathbf{X})$ and the softmax vector: $\mathbf{v} = \sigma(\mathbf{z})$.

**Step 2:** Apply $M$ transforms to the original input image $\mathbf{X}$ to obtain its transformed images $\mathbf{X}^{(i)}$ ($i \in \mathcal{I}_M$), then obtain their corresponding softmax vectors: $\mathbf{v}_i = \sigma(f'(\mathbf{X}^{(i)}))$.

**Step 3:** Get the indices $\mathbf{q}$ of the top-$k$ maximum softmax scores from the logit vector $\mathbf{v}$ using Eq. (16).

**Step 4:** Use the generated indices $\mathbf{q}$ to form new vectors $\mathbf{v}_i \in \mathbb{R}^k$, concatenate these new $k$-dimensional vectors resulting in $\oplus_{i \in \mathcal{I}_M} \mathbf{v}_i[\mathbf{q}] \in \mathbb{R}^{M \times k}$, then pass this resulting matrix to the calibrator $g$ to produce the temperature $\tau$ via Eq. (17).

**Step 5:** Apply the learned temperature $\tau$ to the original logit vector and obtain its maximum softmax score via Eq. (18).

**Step 6:** Plug the updated maximum softmax prediction score from Step 5, the ground truth label $y$, and the predicted label $\hat{y}$ into our proposed Correctness-Aware Loss via Eq. (19).

**Return:** Calibrator model weights $\theta$.

---

## B  FURTHER DISCUSSION

**How could the CA loss improve the ECE metric?** ECE bins confidence and calculates the difference between confidence and accuracy of samples in each bin. In the extreme case where bin size is infinitely small, each bin will contain only one sample (assuming no image duplicates), meaning accuracy of each bin is either 100% or 0%. In this scenario, the CA loss will push correct (wrong) predictions to the high (low) confidence, which always reduces ECE. When bin size gradually becomes larger, improvement brought by CA loss will be less definite but still visible.

**Sample-adaptive temperature** used in Joy et al. (2023); Balanya et al. (2022); Wang et al. (2023) and our method has different properties from global temperature Guo et al. (2017). Because global temperature does not change the order of samples ranked by their confidence, it cannot improve the ability of confidence to separate correct and wrong predictions. Sample-adaptive temperature at least has such potential (but under specific design). On the other hand, trained with the CE loss, the sample-adaptive temperature is demonstrated to produce a competitive calibrator for IND test sets Joy et al. (2023); Balanya et al. (2022). But issues with CE and the lack of using additional information limit its effectiveness for OOD data. In comparison, our method is competitive on both IND and OOD test sets.

**Why the CA loss sometimes still have empirical failures?** A calibrator perfectly optimized by the CA loss will give 0 ECE, because all the correctly (wrongly) classified samples will have confidence of 1 (0). In practice, however, the bottleneck is to tell prediction correctness. We use augmented images but it might not be an optimal solution. In future we will explore new methods for correctness prediction.

**Correctness prediction performance as a potential calibration metric.** Given a confidence value, better separability between correct and wrong predictions leads to safer decision making for users, because less mistakes are made. This paper uses area under the ROC curve (AUROC) to measure the performance of predicting classification correctness. As shown in Sec. D.2, using a calibration method does not always mean user makes less mistakes during decision making. Considering the strong tie between this 2-way classification problem and model calibration (Eq. (1)), we think AUROC can be an additional evaluation metric for model calibration.

**Impact of $k$.** We use the indices of top-$k$ confidences to locate and select the $k$-dim Softmax vectors from the transformed images. In Fig. 6, we find that for various values of $k$ our method improves over uncalibrated models. Moreover, $k > 5$ does not bring much improvement. Considering computational cost, we use $k = 4$. Note $k$ is chosen on the ImageNet-A test set and applied on all the other test sets.

**Computational cost.** On a server with 1 GeForce RTX 3090 GPU, it takes our method 583 seconds to train a calibrator on ImageNet-Val; in comparison, it take PTS and temperature scaling 987 seconds and 32 seconds respectively in training. Because temperature scaling only learns a single parameter (*i.e.*, temperature), it is the quickest to train. The inference time for ours, temperature scaling, and PTS is similar: 2.33, 1.63, and 2.8 milliseconds per image, respectively. The time complexity of our method is the same as that of PTS.

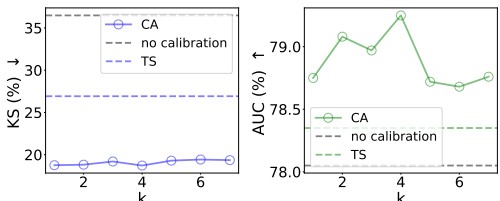

Figure 6: **Impact of $k$ in top-$k$ index selection (Sec. 3.3).** We use ObjectNet test set. Under various $k$ our method is better (lower KS and higher AUROC) than uncalibrated models and TS. We choose $k = 4$ as trade-off between performance and computational cost.

## C  ACCESS OF BENCHMARKS AND MODELS

In this section, we introduce the benchmarking datasets and classification models used in our paper.

**ImageNet models.** We employ the ImageNet models from the PyTorch Image Models (timm) library (Wightman, 2019), which offers models trained or fine-tuned on the ImageNet-1k training set (Deng et al., 2009). The models utilized in our paper are listed below:

{ *'beit_base_patch16_384', 'tv_resnet152', 'tv_resnet50', 'tv_resnet101', 'densenet121', 'inception_v4', 'densenet201', 'vit_base_patch16_384', 'deit_base_patch16_224', 'inception_v3'* }

**Datasets.** We present the test sets employed in the main paper to evaluate the aforementioned ImageNet models. Datasets mentioned below can be accessed publicly via the provided links.

*ImageNet-A(dversarial)* (Hendrycks et al., 2021b): https://github.com/hendrycks/natural-adv-examples.
*ImageNet-S(ketch)* (Wang et al., 2019): https://github.com/HaohanWang/ImageNet-Sketch.
*ImageNet-R(endition)* (Hendrycks et al., 2021a): https://github.com/hendrycks/imagenet-r.
*ImageNet-Blur* (Hendrycks & Dietterich, 2019): https://github.com/hendrycks/robustness.
*ObjectNet* (Barbu et al., 2019): https://objectnet.dev/download.html.

**CIFAR-10 models.** We employ the CIFAR-10 models from the open source library (https://github.com/kuangliu/pytorch-cifar) which offers models trained or fine-tuned on the CIFAR-10 training set (Krizhevsky et al., 2009). The models utilized in our paper are listed below:

{ *'VGG19', 'DenseNet121', 'DenseNet201', 'ResNet18', 'ResNet50', 'ShuffleNetV2', 'MobileNet', 'PreActResNet101', 'RegNetX_200MF', 'ResNeXt29_2x64d'* }

**Datasets.** Datasets used in the CIFAR10 setup can be found through the following links. *CINIC* (Darlow et al., 2018):
https://github.com/BayesWatch/cinic-10.    *CIFAR10-C* Hendrycks & Dietterich (2019)(https://github.com/hendrycks/robustness);

**iWildCam models.** We use the iWildCam models from the open source library (https://worksheets.codalab.org/worksheets/0x52cea64d1d3f4fa89de326b4e31aa50a) which offers models trained/fine-tuned on the iWildCam training set (Beery et al., 2020). The models utilized in our paper are listed below:

{ *'iwildcam_erm_seed1', 'iwildcam_deepCORAL_seed0', 'iwildcam_groupDRO_seed0', 'iwildcam_irm_seed0','iwildcam_erm_tune0', 'iwildcam_ermaugment_tune0', 'iwildcam_ermoracle_extraunlabeled_tune0', 'iwildcam_swav30_ermaugment_seed0', 'iwildcam_dann_coarse_extraunlabeled_tune0', 'iwildcam_afn_extraunlabeled_tune0'* }

**Dataset.** *iWildCam-OOD* (Beery et al., 2020) can be download from the the official guidance: https://github.com/p-lambda/wilds/.

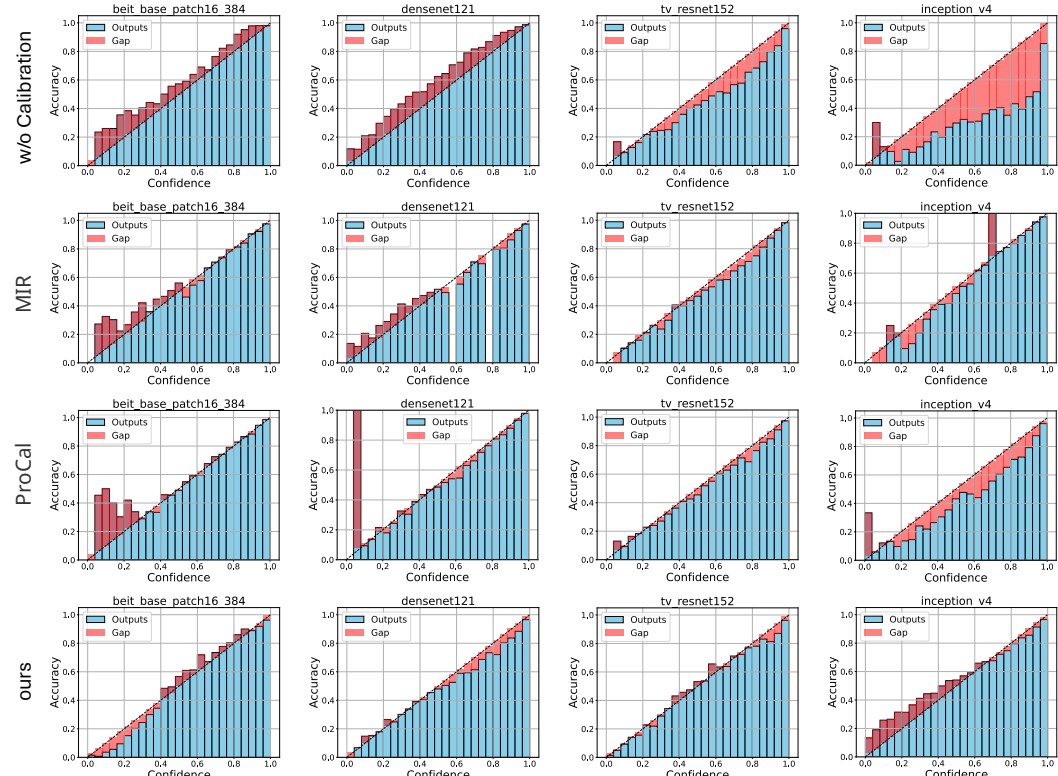

Figure 7: **Reliability diagram of various models on the ImageNet validation set (25 bins)**. Bars above the dashed line indicate underconfidence, while those below indicate overconfidence. Our method effectively mitigates both overconfident and underconfident predictions across different scenarios.

# D ADDITIONAL VISUALISATIONS

## D.1 RELIABILITY DIAGRAM

## D.2 AUROC CURVES

Fig. 9 shows the comparison of Receiver Operating Characteristic (ROC) curves across different calibration methods.

## D.3 DISTRIBUTIONS OF PREDICTIONS

Visualizations of the distributions for correct and incorrect predictions on four datasets are given in Fig. 10.

## LIMITATION

Our method is limited in deployment environments where computational resources are extremely constrained because it requires predicting results from transformed images.

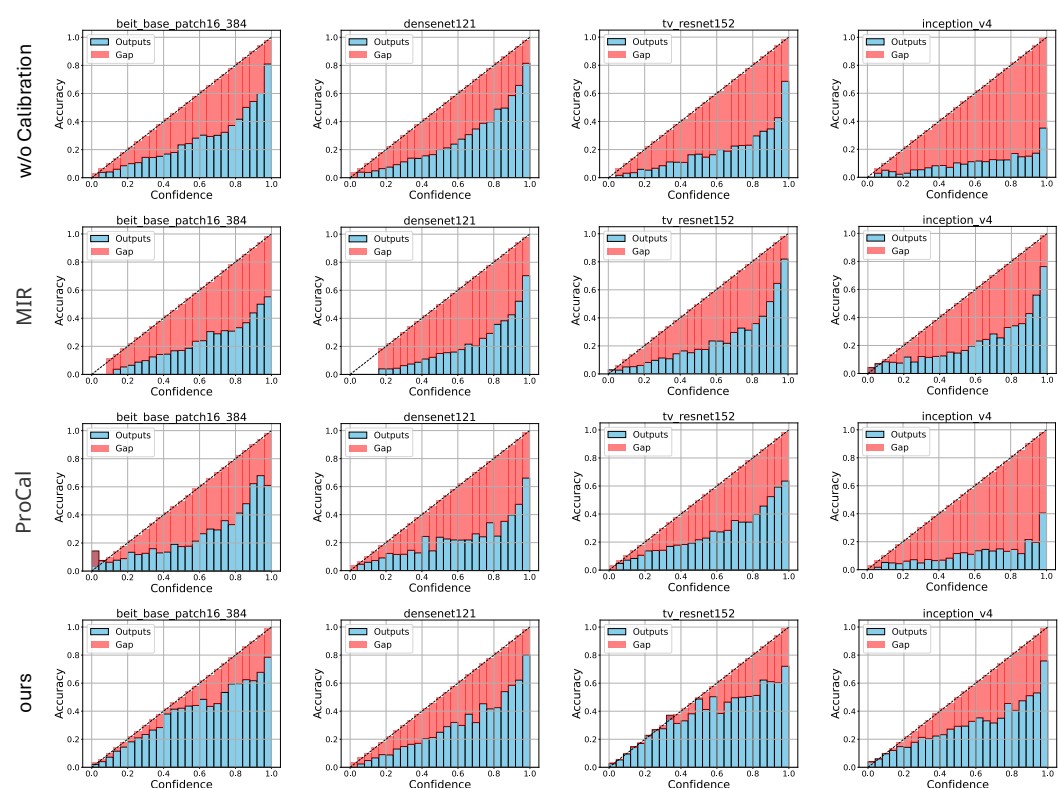

Figure 8: **Reliability diagram of various models on the out-of-distribution (OOD) ObjectNet dataset (25 bins)**. Bars above the dashed line indicate underconfidence, while those below indicate overconfidence. ObjectNet is a highly challenging OOD test set, where models often exhibit severe overconfidence. Our method significantly mitigates this issue.

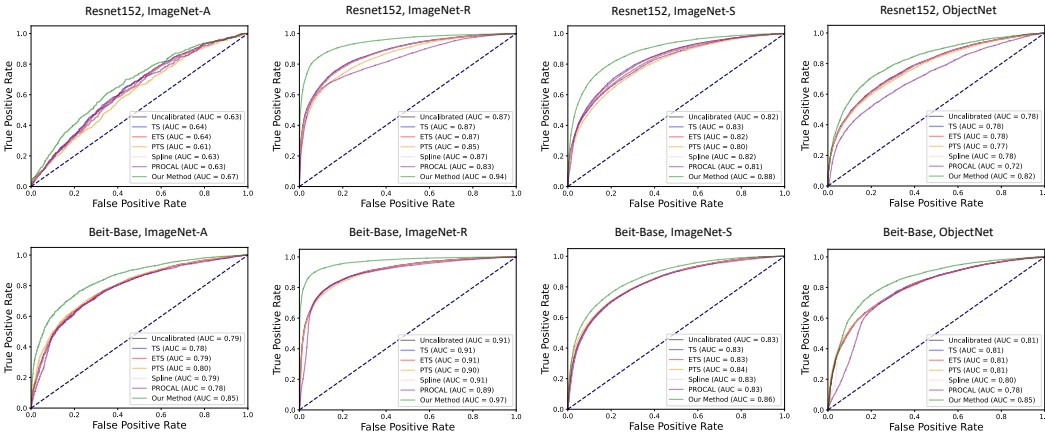

Figure 9: **Comparison of Receiver Operating Characteristic (ROC) Curves Across Different Calibration Methods**. Each figure's title specifies the classifier and the test set used. It is evident that our methods (green curves) yield a higher area under ROC curve (AUROC) compared to other calibration methods, signifying an enhanced ability of our model to distinguish between correct and incorrect predictions based on calibrated confidence.

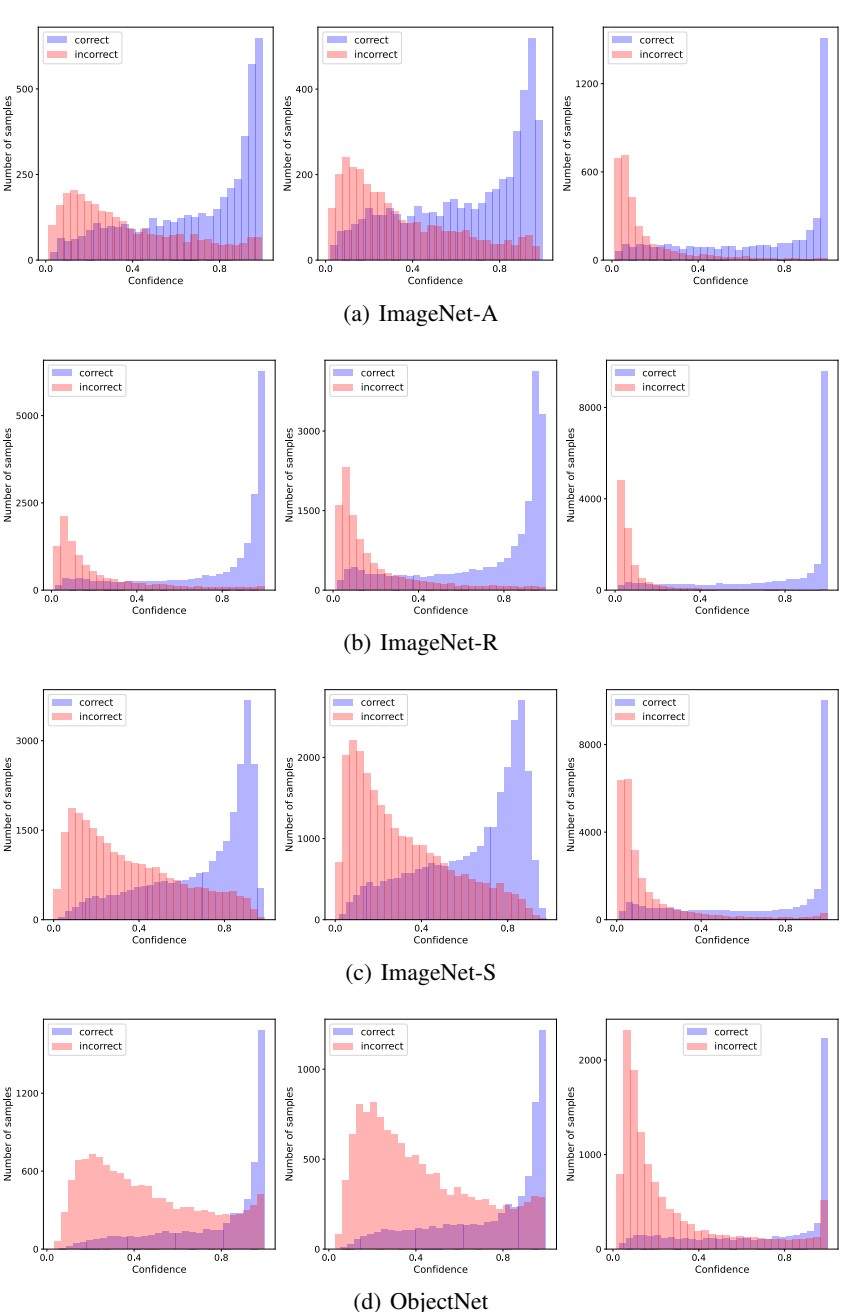

(a) ImageNet-A

(b) ImageNet-R

(c) ImageNet-S

(d) ObjectNet

Figure 10: Visualization of the distributions for correct and incorrect predictions of 'beit_base_patch16_384' on (a) ImageNet-A, (b) ImageNet-R, (c) ImageNet-S, and (d) Object-Net. From left to right, the methods are no calibration, temperature scaling, and our method. We find that our method can better distinguish between correct and incorrect predictions by increasing the confidence value for correct predictions and decreasing it for incorrect ones.

