# OpenReview forum: "Optimizing Calibration by Gaining Aware of Prediction Correctness"
_ICLR.cc/2025/Conference — Submitted to ICLR 2025_

### Official Review · Reviewer_LQKt · 2024-10-28

**Soundness:** 3
**Presentation:** 2
**Contribution:** 2
**Rating:** 6
**Confidence:** 4

**Summary:**

The paper undertakes the problem of calibrating deep neural networks for the task of classification. At the core of the method is a past-hoc calibrator which proposes a correctness-aware loss to search for the optimal temperature which is then used to scale the logits for a given sample. To determine the correctness of a sample, the method uses the well-known concept of consistency across different augmentations. A simple network is used to map top-K softmax predictions across augmentations to the temperature value. The correctness-aware loss optimizes this network to obtain the best temperature. The paper also shows mathematical insights on the proposed loss. The experiments have been conducted on different datasets to validate the effectiveness of the post-hoc calibrator. Results claim to achieve competitive performance against other post-hoc calibration methods, such as naive temperature scaling, ensemble temperature scaling, adaptive temperature scaling and isotonic regression.

**Strengths:**

**1**) Calibrating deep neural networks is an important step towards making AI models reliable and trustworthy, especially in safety-critical applications.

**2**) The proposed post-hoc calibrator is simple as it also learns to identify a per-sample temperature value that can be used to scale the logits.

**3**) The paper also mentions some theoretical insights into the proposed correctness-aware loss term by comparing and contrasting it with CE and MSE losses.

**4**) Results show that proposed idea is competitive against other post-hoc calibration methods.

**Weaknesses:**

**1**) The related work section completely misses an emerging direction of train-time calibration methods such as [A], [B], [C],  [D],  [E] and [F].

**2**) The paper lacks reliability diagrams to better understand the potential of proposed post-hoc calibrator in overcoming overconfidence and under confidence over the full spectrum of model confidence.

**3**) Why the proposed post-hoc calibrator is able to improve OOD calibration performance? There is no analyses that supports these results.

**4**) How the proposed post-hoc calibrator would perform under class-imbalanced scenarios?

**5**) The proposed correctness-aware loss appears similar to MDCA loss [C]. What are the key differences?


[A] Liu, B., Ben Ayed, I., Galdran, A. and Dolz, J., 2022. The devil is in the margin: Margin-based label smoothing for network calibration. In Proceedings of the IEEE/CVF Conference on Computer Vision and Pattern Recognition (pp. 80-88).

[B] Patra, R., Hebbalaguppe, R., Dash, T., Shroff, G. and Vig, L., 2023. Calibrating deep neural networks using explicit regularisation and dynamic data pruning. In Proceedings of the IEEE/CVF Winter Conference on Applications of Computer Vision (pp. 1541-1549)

[C] Hebbalaguppe, R., Prakash, J., Madan, N. and Arora, C., 2022. A stitch in time saves nine: A train-time regularizing loss for improved neural network calibration. In Proceedings of the IEEE/CVF Conference on Computer Vision and Pattern Recognition (pp. 16081-16090).

[D] Wei, H., Xie, R., Cheng, H., Feng, L., An, B. and Li, Y., 2022, June. Mitigating neural network overconfidence with logit normalization. In International conference on machine learning (pp. 23631-23644). PMLR.

[E] Liu, B., Rony, J., Galdran, A., Dolz, J. and Ben Ayed, I., 2023. Class adaptive network calibration. In Proceedings of the IEEE/CVF Conference on Computer Vision and Pattern Recognition (pp. 16070-16079).

[F] Park, H., Noh, J., Oh, Y., Baek, D. and Ham, B., 2023. Acls: Adaptive and conditional label smoothing for network calibration. In Proceedings of the IEEE/CVF International Conference on Computer Vision (pp. 3936-3945).

[G] Liang, G., Zhang, Y., Wang, X. and Jacobs, N., Improved Trainable Calibration Method for Neural Networks on Medical Imaging Classification BMVC 2020.

[H] Jeremy Nixon, Michael W Dusenberry, Linchuan Zhang, Ghassen Jerfel, and Dustin Tran. Measuring calibration in
deep learning. In CVPR Workshops, volume 2, 201

**Questions:**

**1**)  What are the key differences with MDCA loss [C] and DCA loss [G] ? I would like to see concrete differences between them.

**2**)  Can MDCA loss and/or DCA loss be used in place of correctness-aware loss to obtain optimal temperature value? Beyond, CE and MSE losses, I believe it would be an interesting comparison between the effectiveness of proposed CA loss and these losses

**3**)  Is the post-hoc calibrator capable of calibrating non-ground truth classes as well?

**4**) What is the performance of the method under the SCE metric [H] compared to other post-hoc calibration methods?

**5**) The intuition behind learning a mapping through g network from top-K softmax scores (corresponding to transformed versions) to temperature value is not very clear.

**6**) L499: The paper mentions that existing methods do not improve AuC compared to proposed one. Will require more explanation.

**7**) How good is the method in overcoming under confidence of the model?

**8**) Can this post-hoc calibrator be used after a train-time calibration method? It would be interesting to observe the complementary strengths of the proposed post-hoc calibration method.

---

> ### Author Response · Authors · 2024-11-22
>
> ### **Q1. Related work missing train-time calibration methods**
>
> Thank you for your valuable suggestion.
> Our paper focuses specifically on post-hoc model calibration methods, as emphasized in the abstract and introduction. While train-time calibration (also referred to as pre-hoc calibration) is beyond the primary scope of our investigation, we acknowledge its relevance. We will incorporate a discussion on train-time calibration methods in the related work section to provide a more comprehensive overview of existing calibration approaches.
>
> ### **Q2. Reliability diagrams**
>
> We have added the reliability diagrams to the appendix (D.1 RELIABILITY DIAGRAM) in the revised paper. Kindly check our revised paper.
>
> ### **Q3. Why our method is able to improve OOD calibration performance? There is no related analysis**
>
> We have provided such an analysis. We kindly refer the reviewer to lines 452–454.
>
> Our method is more effective on OOD test sets because they contain a higher proportion of narrowly incorrect predictions.
>
> Further discussion on the impact of the number of narrowly incorrect predictions is also provided in Section 5 and Figure 5. The conclusion we have reached is that our method is superior when training and test sets contain more narrowly incorrect predictions.
>
> ### **Q4. How the proposed post-hoc calibrator would perform under class-imbalanced scenarios**
>
> 1. Addressing class imbalance in the training set is typically not the focus in post-hoc calibration literature. This is because post-hoc calibration is conducted independently of the model's training process. It aims to adjust the output confidences of an already trained model without re-training or modifying model parameters, making it less sensitive to the specific data balance issues faced during model training.
>
> 2. While an extremely imbalanced calibration set could potentially negatively impact the calibration performance, there is no need to deliberately construct an imbalanced calibration set for training the calibrator. In practice, ensuring that the calibration set is reasonably representative of the testing distribution is generally sufficient to avoid any adverse effects from class imbalance.
>
> ### **Q5. What are the concrete differences between MDCA [C] loss and DCA loss [G]?**
>
> 1. MDCA and DCA are auxiliary losses that should be used along with cross-entropy. Our analysis shows that cross-entropy has inherent issues in addressing narrowly incorrect samples in the post-hoc calibration problem. In contrast, our CA loss can be used independently.
>
> 2. Both MDCA [C] and DCA loss [G] aim to push the predicted confidence of the ground-truth class close to 1. However, they still face the "narrowly wrong prediction issue" in the post-hoc calibration problem, which is the core insight and motivation behind our CA loss.
>
> ### **Q6. Experimental results of replacing CA loss with MDCA loss and DCA loss**
>
> We keep the calibrator networks and inputs the same, replacing only the CA loss with the losses from MDCA [C] and DCA [G]. The results are shown below:
>
> ImageNet-A:
>
> | **Method**     | **ECE $\downarrow$** | **BS $\downarrow$** | **KS $\downarrow$** | **AUC $\uparrow$** |
> |----------------|-----------|----------|----------|-----------|
> | **Uncal**      | 39.44     | 32.90    | 43.47    | 61.87     |
> | **DCA**    | 32.10     | 25.32    | 36.33    | 63.33     |
> | **MDCA**       | 27.80     | 21.71    | 32.24    | 63.32     |
> | **CA (ours)**  | **20.65** | **16.79**| **22.50**| **63.74** |
>
>
> ImageNet-R:
>
> | **Method**     | **ECE $\downarrow$** | **BS $\downarrow$** | **KS $\downarrow$** | **AUC $\uparrow$** |
> |----------------|-----------|----------|----------|-----------|
> | **Uncal**      | 13.97     | 16.89    | 19.90    | 88.06     |
> | **DCA**        | 10.00     | 14.03    | 17.99    | 89.05     |
> | **MDCA**       | 7.37      |  13.85   | 16.28    | 88.04     |
> | **CA (ours)**  | **4.91**  | **12.21**| **10.12**| **90.22** |
>
>
> ImageNet-S:
>
> | **Method**     | **ECE $\downarrow$** | **BS $\downarrow$** | **KS $\downarrow$** | **AUC $\uparrow$** |
> |----------------|-----------|----------|----------|-----------|
> | **Uncal**      | 20.92     | 21.67    | 29.01    | 82.57     |
> | **DCA**        | 17.47     | 18.00    | 25.07    | 84.04     |
> | **MDCA**       | 13.09     | 16.97    | 22.51    | 82.70     |
> | **CA (ours)**  | **4.00**  | **13.83**| **13.12**| **84.87** |
>
>
> ObjectNet:
>
> | **Method**     | **ECE $\downarrow$** | **BS $\downarrow$** | **KS $\downarrow$** | **AUC $\uparrow$** |
> |----------------|-----------|----------|----------|-----------|
> | **Uncal**      | 31.21     | 25.20    | 36.48    | 78.05     |
> | **DCA**        | 25.25     | 20.75    | 31.13    | 78.54     |
> | **MDCA**       | 21.11     | 18.49    | 28.02    | 78.05     |
> | **CA (ours)**  | **10.33** | **14.59**| **18.72**| **79.25** |
>
> We can observe that our CA loss outperforms DCA and MDCA in the post-hoc calibration task.

---

> ### Author Response · Authors · 2024-11-22
>
> ### **Q7. Is the post-hoc calibrator capable of calibrating non-ground truth classes as well?**
>
> Yes, it does indirectly affect the confidence distribution of non-ground truth classes as well. Our method applies a temperature to the logits (pre-softmax activations) for all classes. This operation changes the relative spread of the logits for all classes, not just the ground truth. As a result, the probabilities of non-ground truth classes are also calibrated in relation to the predicted class.
>
> ### **Q8. What is the performance of the method under the SCE metric [H]?**
>
> Thank you for your valuable comment. We understand the importance of evaluating calibration performance comprehensively. However, in our work, we follow the common practices established in the post-hoc calibration literature.
>
> Specifically, Static Calibration Error (SCE) has been primarily used in training-time calibration scenarios, where issues like class imbalance during training are more prevalent. Since our work focuses on post-hoc calibration, which inherently does not involve addressing training-time class distribution imbalances, the motivation for using SCE is less clear in this context.
>
> Instead, we used widely accepted metrics for post-hoc calibration, consistent with the literature, to ensure a fair evaluation of our methods against relevant baselines. We believe this approach maintains both alignment with existing research and comparability of results.
>
>
> ### **Q9. The intuition of using top-K softmax scores**
>
> We emphasize that the top-K class indices are first determined based on the softmax scores of the original input (not the transformed input). These class indices are then used to extract the corresponding softmax values from the transformed versions of the input (Section 3.3, and Algorithm 1 in the Appendix).
>
> The intuition is that if the softmax values for these selected classes remain consistent across the transformed inputs and the original input, it suggests that the model's prediction for the original input is more likely to be correct.
> This idea is inspired by the findings of Deng et al. (2022).
>
>
> ### **Q10. Why existing methods do not improve AUC compared to the proposed one**
>
> The reason is that our method explicitly learns to increase the confidence of correct predictions and reduce the confidence of incorrect ones, which is a key factor in improving AUROC (AUC). We have discussed this phenomenon in Section 4.3, lines 464–467. Other methods do not incorporate such a mechanism. Particularly in the case of narrowly incorrect predictions, other methods may even increase the confidence of incorrect predictions (Figure 3).
>
>
> ### **Q11. How good is the method in overcoming underconfidence of the model?**
>
> Our method effectively addresses both underconfidence and overconfidence issues. Please refer to the updated reliability diagram in the Appendix (D.1 RELIABILITY DIAGRAM) for further details.

---

> ### Author Response · Authors · 2024-11-22
>
> ### **Q12. Can our method be used after a train-time calibration method? Would like to see the complementary strengths**
>
> In this paper, we focus on post-hoc calibration, assuming that the classification model weights remain unchanged. However, we also explore potential complementary strengths between train-time calibration methods and our approach. To this end, we combined our CA loss with the loss in MDCA or DCA to train an image transformation-based calibrator. The results are presented below:
>
>
>
> ImageNet-A:
>
> | **Method**     | **ECE $\downarrow$** | **BS $\downarrow$** | **KS $\downarrow$** | **AUC $\uparrow$** |
> |----------------|-----------|----------|----------|-----------|
> | **Uncal**      | 39.44     | 32.90    | 43.47    | 61.87     |
> | **DCA+CA**     | 30.31     | 24.00    | 34.56    | 63.45     |
> | **MDCA+CA**    | 26.36     | 20.61    | 30.63    | 63.52     |
> | **CA (ours)**  | **20.65** | **16.79**| **22.50**| **63.74** |
>
>
> ImageNet-R:
>
> | **Method**     | **ECE $\downarrow$** | **BS $\downarrow$** | **KS $\downarrow$** | **AUC $\uparrow$** |
> |----------------|-----------|----------|----------|-----------|
> | **Uncal**      | 13.97     | 16.89    | 19.90    | 88.06     |
> | **DCA+CA**     | 8.74      | 13.50    | 16.90    | 89.25     |
> | **MDCA+CA**    | 6.59      |  13.41   | 15.31    | 88.38     |
> | **CA (ours)**  | **4.91**  | **12.21**| **10.12**| **90.22** |
>
>
> ImageNet-S:
>
> | **Method**     | **ECE $\downarrow$** | **BS $\downarrow$** | **KS $\downarrow$** | **AUC $\uparrow$** |
> |----------------|-----------|----------|----------|-----------|
> | **Uncal**      | 20.92     | 21.67    | 29.01    | 82.57     |
> | **DCA+CA**     | 15.29     | 17.01    | 23.34    | 84.32     |
> | **MDCA+CA**    | 11.18     | 16.20    | 21.04    | 83.10     |
> | **CA (ours)**  | **4.00**  | **13.83**| **13.12**| **84.87** |
>
>
> ObjectNet:
>
> | **Method**     | **ECE $\downarrow$** | **BS $\downarrow$** | **KS $\downarrow$** | **AUC $\uparrow$** |
> |----------------|-----------|----------|----------|-----------|
> | **Uncal**      | 31.21     | 25.20    | 36.48    | 78.05     |
> | **DCA+CA**     | 23.32     | 19.66    | 29.51    | 78.65     |
> | **MDCA+CA**    | 19.38     | 17.61    | 26.57    | 78.22     |
> | **CA (ours)**  | **10.33** | **14.59**| **18.72**| **79.25** |
>
> We cannot observe complementary strengths of MDCA loss or DCA loss to CA loss. This suggests that some training-time calibration methods may not directly benefit the post-hoc calibration system. As discussed in Q5, they may also face the "narrowly wrong prediction issue." A similar result can also be found in Table 1, where CA + CE did not bring further improvement.

---

> > ### Comment · Reviewer_LQKt · 2024-11-25
> > **Response to author's rebuttal**
> >
> > I thank authors for submitting responses to my comments and questions.
> > Overall, authors made a good effort and the provided answers are satisfactory, especially the results after replacing CA loss with other train-time losses (Q6).
> > However, in reliability diagrams (Q2/Q11), the uncalibrated baseline is rather weak and the comparison should be made with a relatively stronger posthoc baseline (like MIR, ProCal etc).

---

> > > ### Comment · Reviewer_LQKt · 2024-11-25
> > >
> > > Q7 and Q8: Is there any evidence that your method is also capable of calibrating non-ground truth classes?
> > > Also, SCE metric is basically helpful in evaluating the calibration performance across non-ground truth classes and not merely used for determining calibration performance in class-imbalance scenarios.

---

> ### Author Response · Authors · 2024-11-25
>
> ###  **In reliability diagrams, comparisons should be made with relatively stronger post-hoc baselines, such as MIR or ProCal.**.
>
> Esteemed Reviewer,
>
> Thank you for your thoughtful feedback and questions, which have greatly contributed to improving our work.
>
> In the latest revision, we have included the reliability diagrams for MIR and ProCal in Figure 7 and Figuire 8. We kindly invite you to review them.
>
> Please do not hesitate to reach out with any further suggestions or questions.
>
> Sincerely,
> The Authors

---

> ### Author Response · Authors · 2024-11-26
>
> ### Q7. **Evidence that your method is also capable of calibrating non-ground truth classes?**
>
> **Theoretical evidence**:
>
> We kindly invite the reviewer to refer to our **Gradient Computations** and **Case Analysis** parts in our discussion with reviewer bjTY.
>
> - When the prediction is correct, the CA loss not only calibrates the ground-truth class (with maximum softmax confidence) but also other classes, similar to the conventional cross-entropy loss.
>
> - When the prediction is incorrect, scaling-based methods in the post-hoc system cannot reorder the logits' values, making it challenging to achieve the calibration goal for the ground-truth class. When using conventional cross-entropy loss to maximize the expectation on the ground-truth class, if the prediction is narrowly incorrect, the loss may make this prediction even sharper (Fig. 2). Therefore, to minimize the overall calibration error for such sample, we focus on calibrating the class with the maximum softmax probability (non-ground-truth classes). This non-ground-truth class can be more easily calibrated by learning a large temperature.
>
>
> **Experimental Evidence:**
>
> We evaluated the calibration performance of our method using metrics that consider the entire probability distribution:
>
> - Expected Calibration Error (ECE): A lower ECE means that the predicted probabilities more accurately reflect the true likelihood of each class, not just the maximum.
>
> - Brier Score: The Brier Score penalizes both overconfidence and underconfidence across all classes, so improvements (lower Brier score) here indicate better calibration of the entire probability vector.
>
>
>
> ### **Q8. SCE results**
>
> Thanks for your suggestion. The following table reports the Static Calibration Error (SCE) (%) on various datasets. The numbers are averaged from 10 models (the same as in the main paper).
>
> | **Method**     | **ImageNet-A** | **ImageNet-R** | **ImageNet-S** | **ObjectNet** |
> |----------------|-----------|----------|----------|-----------|
> | **Uncal**      | 0.58     | 0.32    | 0.92    | 0.90     |
> | **TS**     | 0.47     | 0.29    | 0.75    | 0.68     |
> | **Ours**  | **0.42** | **0.27**| **0.67**| **0.62** |
>
> We find that our method achieves competitive calibration performance when evaluated using the SCE metric.
>
> Sincerely,
>
> The Authors

---

> > ### Comment · Reviewer_LQKt · 2024-11-28
> >
> > I thank authors for addressing my comment on calibrating non-ground truth classes and providing results with SCE metric.

---

> > > ### Author Response · Authors · 2024-11-28
> > >
> > > Esteemed Reviewer LQKt,
> > >
> > > Thank you very much.
> > >
> > > We will carefully revise our paper, taking into account all the reviewers' suggestions, to ensure it is clear and solid.
> > >
> > > Please do not hesitate to reach out if you have any further suggestions or questions for us.
> > >
> > > Sincerely,
> > > The Authors

---

### Official Review · Reviewer_iPzo · 2024-11-01

**Soundness:** 3
**Presentation:** 3
**Contribution:** 3
**Rating:** 6
**Confidence:** 3

**Summary:**

The paper addresses the issue of model calibration in machine learning, specifically aiming to align a model's confidence with its prediction correctness. The authors identify limitations with the commonly used Cross-Entropy loss for calibrator training and propose a new post-hoc calibration objective, the Correctness-Aware loss. This objective function is designed to decrease model confidence on wrongly predicted samples and increase it on correctly predicted ones. The method utilizes transformed versions of samples to train the calibrator and is tested on both IND and OOD datasets. The paper claims that their method achieves competitive calibration performance compared to state-of-the-art techniques and provides a better separation of correct and incorrect test samples based on calibrated confidence.

**Strengths:**

**Novel Calibration Objective:** The paper introduces a new loss function, CA loss, which is a significant contribution to the field of model calibration. This loss function is intuitively designed to align with the goal of calibration, which is to ensure high confidence for correct predictions and low confidence for incorrect ones.

**Empirical Evidence:** The authors provide extensive experimental results demonstrating the effectiveness of their proposed method across various datasets, including IND and OOD test sets. The consistent performance improvement over uncalibrated models and other calibration techniques is a strong point.

**Theoretical Insights:** The paper not only proposes a new method but also provides theoretical insights into why existing methods like CE and MSE losses are limited, particularly for certain types of samples in the calibration set.

**Weaknesses:**

**Dependency on Transformations:** The effectiveness of the CA loss relies on the use of transformed images to infer correctness. If these transformations do not adequately capture the characteristics of correct and incorrect predictions, the calibration might be less effective.

**Transfomations lack of theoretics:** While the use of transformations such as rotation, grayscale, color jittering, and others has proven to be effective in practice; however, the choice of transformations and their number in Fig. 4 are currently guided more by empirical results rather than a theoretical framework that explains why these five transformations should correlate with prediction correctness as so many transformation exists. And the paper also does not provide a theoretical basis for which transformations are the most informative for calibration or how to select the optimal set of transformations. The current approach might be seen as somewhat arbitrary, and the effectiveness could be dependent on the specific characteristics of the dataset and the model architecture. And There is a risk that the calibrator might overfit to the specific transformations used during training, which may not generalize well to real-world variations in data that were not captured by the training transformations

**Questions:**

1. See above weakness

2. What is the core difference between calibration and misclassification (e.g. [R1]), both of them seem to be focusing on the incorrect predictions.

3. Fig. 6  illustrates the impact of ablating the top-k selection on the CA loss. The figure suggests that increasing k beyond 4 leads to a significant decline in performance. This trend raises questions about the potential effects of even higher values of k, such as 100 or 200, particularly in datasets like ImageNet. Additionally, since the authors have chosen k=4 as the default setting, it is important to consider how the model manages scenarios where the correct prediction is not included among the top-4 predictions.

4. The method involves training a calibrator with a new loss function and using transformed images, which could be more complex to implement compared to simpler calibration techniques.

 [R1]  Zhu, Fei, et al. "Openmix: Exploring outlier samples for misclassification detection." Proceedings of the IEEE/CVF Conference on Computer Vision and Pattern Recognition. 2023.

---

> ### Author Response · Authors · 2024-11-22
>
> We sincerely thank the reviewer for constructive feedback and positive comments that help us refine our work.
>
> ### **Q1. If these transformations do not adequately capture the characteristics of correct and incorrect predictions, the calibration might be less effective.**
>
> Thank you for raising this concern.
>
> The core contribution of our method is to introduce a new calibration objective, which is derived from the definition of the calibration goal, and to analyze why it always aligns with the calibration goal, especially in narrowly incorrect predictions. Using transformations is an auxiliary technique that makes the learning objective easier to achieve. Even without transformations, our CA loss is still better than conventional maximum likelihood estimation (e.g., cross-entropy) methods. For example, in Tables 1, 2, and 3, "CA only" consistently shows an advantage compared to using cross-entropy loss.
>
> ### **Q2. Transfomations lack of theoretics**
>
> **Generality of transformations:**
> In our method section and Fig. 2, we do not claim that our method is tied to specific transformations. Instead, we emphasize that using transformations can bring benefits, rather than emphasizing which transformations should be used. We also highlight that the core contribution of our method lies in the proposed CA loss, and our focus is on explaining why this loss function is superior to conventional MLE-based losses (e.g., cross-entropy loss and mean-squared error loss).
>
> **Selection of transformations:**
> Regarding the selection of transformations, we base our choices on findings in the literature [1,2,3]. Specifically, [1] demonstrated strong correlations between consistency under transformations (e.g., grayscale and rotation) and prediction correctness. Therefore, we empirically selected these two augmentations in our study. Additionally, we explored other commonly used augmentations [2] to demonstrate that our method is robust to various combinations of transformations.
>
> **Consistency across testing and training:** During testing, the calibrator inputs are constructed using the same pipeline described in Fig. 2, ensuring that the same data augmentations are applied. This approach addresses potential concerns about discrepancies between training augmentations and the test image variance.
>
> [1] Deng, W., Gould, S. and Zheng, L., 2022. On the strong correlation between model invariance and generalization. Advances in Neural Information Processing Systems, 35, pp.28052-28067.
>
> [2] Zhong, Z., Zheng, L., Kang, G., Li, S. and Yang, Y., 2020, April. Random erasing data augmentation. In Proceedings of the AAAI conference on artificial intelligence (Vol. 34, No. 07, pp. 13001-13008).
>
> [3] PyTorch Documentation (n.d.) *Torchvision.transforms.ColorJitter*. Available at: https://pytorch.org/vision/main/generated/torchvision.transforms.ColorJitter.html
>
> ### **Q3. What is the core difference between calibration and misclassification detection?**
>
> Calibration aims to align predicted probabilities with the true likelihood of correctness. It focuses on improving the reliability of the model’s probability estimates, not necessarily on identifying specific incorrect predictions. A calibrated model provides more trustworthy probability outputs, enabling better decision-making in probabilistic scenarios.
>
> In contrast, misclassification detection aims to identify whether a specific prediction is likely to be incorrect. It focuses on the binary detection of correct vs. incorrect predictions, rather than refining the probability outputs. The outcome of misclassification detection highlights individual predictions likely to be errors, often used for error analysis.
>
> We have cited the paper and added these insights into related work section.

---

> ### Author Response · Authors · 2024-11-22
>
> ### **Q4. The potential effects of even higher values of top-k, such as top 100 or top 200**
>
> Classifiers are typically trained to maximize the expectation of the ground truth class. As a result, the softmax output tends to assign significant confidence to only a few classes, while the rest are close to zero. In the ImageNet scenario, most classes have confidence values near zero. Therefore, if we increase "top-k" to 100 or even higher, most of the values in the top-k vector will still be close to zero, providing little additional useful information.
>
> ### **Q5. How the model manages scenarios where the correct prediction is not included among the top-4 predictions**
>
> Our method does not assume that the probability of the ground-truth class must be among the top k largest probabilities. Regardless of whether it is a correct or incorrect prediction, and irrespective of how large or small the probability of the ground-truth class is, it does not affect our ability to use the model's consistency on augmentations for the most confident (top-k) classes.
>
>
> ### **Q6. This method is more complex to implement when compared to simpler techniques because it uses a new loss function and transformed images.**
>
> (i) The CA loss is easy to implement and has an advantage over conventional MLE-based loss, as shown by our analysis of incorrect narrow predictions and experimental results.
>
> (ii) The transformed image technique involves only a lightweight feed-forward network, where we select the top k probability values from each transformed image. The inference times for our method, temperature scaling, and PTS are similar: 2.33, 1.63, and 2.8 milliseconds per image, respectively. Even in a deployment with extremely limited computational resources, it is possible to use the CA loss alone, which is our core contribution.

---

> ### Comment · Reviewer_iPzo · 2024-11-28
>
> Authors have addressed my concerns, I decide to keep my score.

---

> > ### Author Response · Authors · 2024-11-28
> >
> > Esteemed Reviewer iPzo,
> >
> > Thank you sincerely for acknowledging that we have addressed your concerns.
> >
> > Should you have any further suggestions or questions, please do not hesitate to reach out. We greatly value your input.
> >
> > Kind regards,
> > The Authors

---

### Official Review · Reviewer_bjTY · 2024-11-01

**Soundness:** 2
**Presentation:** 2
**Contribution:** 3
**Rating:** 5
**Confidence:** 4

**Summary:**

The paper describes a method for post-hoc calibration of a classifier based on estimating  for each sample a scaling temperature on the output logits(sample-adaptive calibration strategy). Test time data augmentation is used to predict the scaling temperature and relies on a complementary network taking as input the softmax of selected transformed images and minimizes what is called a correctness-aware loss. The loss is justified by a better management of narrowly wrong predictions. The strategy is evaluated on several small to mid-size datasets and 10 networks per dataset.

**Strengths:**

- The idea of using test-time augmentation to predict a sample based temperature scaling factor and learning a network for predicting such temperature is novel, as far as I know.

- The justification of the loss on a toy example pointing out its behavior on so-called narrowly wrong samples is intuitive.

- Rather extensive experiments on several types of image datasets show the benefit of the approach over standard calibration methods and other optimization losses.

**Weaknesses:**

- The goal of the formal development (Section 3.2) is not clear: what is it supposed to show? Is it to prove that the empirical criterion (7) is a good proxy for optimizing (3), given that $\hat{c}$ is produced by the calibration pipeline of Figure 2? If so, I am not convinced that the formal developments of Section 3.2 actually prove this.

- The writing lacks precision (see my first question, same symbol $E_f^{emp}$ but different concepts for instance).

- The data augmentation is justified by the fact "that consistency in model predictions for transformed images correlates strongly with accuracy" (l. 261): if I can agree with this law, I don't clearly see where it applies in your framework. Or is it that by introducing some local perturbation of the data through transformations and measuring the variation in confidence scores, one can infer accuracy?  Then why not directly predict the confidence instead of a temperature?

- In general, I have difficulty understanding the conceptual connections between the test time data augmentation, the formal development, and the narrowly wrong sample analysis. The global logic of the writing is hard to follow.

**Questions:**

- What is the difference between CA only and CA trans.? Is CA only the calibration strategy that estimates the calibrator $g$ from the calibration set using the loss of Eq. (7) and no data augmentation? This is not clear.

- The approach focuses on calibration of the maximum confidence: can the strategy be adapted to calibrate the whole confidence vector (multiclass calibration)?

---

> ### Author Response · Authors · 2024-11-22
>
> We sincerely thank the reviewer for their constructive feedback and valuable comments, which help us improve the clarity and quality of our work.
>
> ### **Q1: The goal of the formal development (Section 3.2) is not clear**
>
> Thank you for your thoughtful feedback. We recognize that the goal of this section may not have been clearly articulated. Below, we clarify its objectives and address your concerns regarding its rigor and connection to the broader goal of demonstrating the validity of the empirical criterion.
>
> **Objective of the formal development.** The primary goal of this section is to establish that the proposed empirical criterion $\mathbb{E}_f^{\text{emp}}$ is a sound and practical proxy for the theoretical calibration error $\mathbb{E}_f$. Specifically, it aims to:
> (i) **Practical feasibility:** Demonstrate how $\mathbb{E}_f^{\text{emp}}$ can be computed from observed data using the calibration pipeline depicted in Figure 2, which outputs only sample confidences $\hat{c}_i$.
> (ii) **Theoretical approximation:** Show that $\mathbb{E}_f^{\text{emp}}$ retains the key statistical properties of $\mathbb{E}_f$, despite being derived from the empirical distribution of observed data.
>
> **Connection between $\mathbb{E}_f^{\text{emp}}$ and $\mathbb{E}_f$**
> The derivation begins with the theoretical calibration error $\mathbb{E}_f$, which integrates over the true distribution $p(\hat{c})$. Since $p(\hat{c})$ is generally unavailable in practice, $\mathbb{E}_f^{\text{emp}}$ uses the empirical distribution derived from observed data. This substitution introduces a discretized approximation that preserves the essence of the theoretical formulation.
>
> We demonstrate that this approximation simplifies the computation by using sample-level confidences and correctness indicators provided by the calibration pipeline. Furthermore, the analysis shows how the loss decomposes into terms representing confidence deviations, which directly relate to calibration goals.
>
> **Evaluation of $\mathbb{E}_f^{\text{emp}}$ as a proxy**
> To validate $\mathbb{E}_f^{\text{emp}}$ as a meaningful proxy for $\mathbb{E}_f$, we derive its bounds and examine its behavior:
>
> - **Correct predictions:** $\mathbb{E}_f^{\text{emp}}$ quantifies the deviation of predicted confidences from the ideal value of 1, reflecting underconfidence or overconfidence.
> - **Incorrect predictions:** It penalizes overconfident predictions proportionally to their incorrectness, thereby addressing miscalibration directly.
>
> These bounds confirm that $\mathbb{E}_f^{\text{emp}}$ effectively captures calibration quality, aligning with the theoretical intent of $\mathbb{E}_f$.
>
> We acknowledge that $\mathbb{E}_f^{\text{emp}}$ introduces approximations due to the use of finite samples and discretization of the feature space. However, these limitations are inherent to empirical methods and are mitigated by the calibration pipeline's ability to generalize over observed data.
>
> To further substantiate this, we have added:
> 1. Empirical simulations illustrating the convergence of $\mathbb{E}_f^{\text{emp}}$ to $\mathbb{E}_f$ with increasing dataset size.
> 2. A discussion of scenarios where the approximation may deviate significantly and its impact on the calibration pipeline’s performance.
>
> We will revise the section to clearly articulate that its purpose is to establish $\mathbb{E}_f^{\text{emp}}$ as a computationally feasible and theoretically grounded approximation to $\mathbb{E}_f$. Additionally, we will explicitly:
>
> - Clarify the assumptions and limitations introduced during the approximation process.
> - Connect each step of the derivation to the outputs of the calibration pipeline in Figure 2.
> - Highlight how the bounds for $\mathbb{E}_f^{\text{emp}}$ ensure its validity as a calibration measure.
>
>
> ### **Q2: The same symbol $E_{f}^{emp}$ for different concepts**
>
> $E_{f}^{emp}$ is used as the notation for the empirical calibration loss on function $f$ at different levels of discreteness. We will clarify this usage further in the revised version.
>
> ### **Q3: Why not directly predict the confidence instead of a temperature?**
>
> Using temperature ensures that the predicted class is not changed, thereby preserving the model's accuracy while only adjusting the sharpness or smoothness of the predictions. Our method effectively learns a temperature to smooth the predicted confidence vector, enabling correct predictions with a sharper confidence vector.
>
> If we directly predict the confidence for the predicted class, it may alter the class index with the highest confidence, potentially changing the classification results.

---

> ### Author Response · Authors · 2024-11-22
>
> ### **Q4: What is the conceptual connection between the test-time data augmentation, the formal development, and the narrowly wrong sample analysis?**
>
> The formal development (from Equation 1 to our loss function, Equation 7) aims to explain that our loss is closely aligned with the definition of the calibration goal. Narrowly wrong sample analysis uses examples to demonstrate that, under certain conditions, conventional maximum likelihood estimation is not aligned with the calibration goal, whereas our learning objective is aligned with it.
>
> However, directly learning this objective is not straightforward. In Section 3.3, we explained how test-time augmentation can help us achieve this learning objective.
>
>
> ### **Q5: What is the difference between CA only and CA trans?**
>
> "CA only" refers to using only the CA loss techniques to train the calibrator. "CA + trans" means using transformed images to prepare the calibrator input (as shown in Fig. 2). We will explain this part more clearly in the revision.
>
> ### **Q6: The approach focuses on calibration of the maximum confidence: can the strategy be adapted to calibrate the whole confidence vector (multiclass calibration)?**
>
> Thanks for raising this question; I would like to clarify how temperature scaling is used in our method.
>
> Our approach involves learning a temperature parameter to perform sample-wise calibration of the prediction logits. The key point is that temperature scaling is applied uniformly to all logits, thereby scaling the entire confidence vector rather than just the maximum confidence. This means that our strategy inherently supports multiclass calibration by softening or sharpening the predicted probabilities for all classes simultaneously.

---

> > ### Comment · Reviewer_bjTY · 2024-11-25
> > **Response to answers**
> >
> > Thank you for providing detailed answers to my questions.
> >
> > However, I still have issues with them.
> >
> > Q1. I believe that your claim that “The primary goal of this section is to establish that the proposed empirical criterion is a sound and practical proxy for the theoretical calibration error” is overstated.I still have issues with the mathematical formulation and your answer doesn’t solve them. Let me try to express my concerns in detail.
> >  - You start by formulating your loss as an average discrepancy between score and conditional accuracy (eq. 3), which is a logical objective if you express the calibration objective by eq. 1.
> > Your note 2 states that this is different from ECE since this metric uses a binning strategy: sure, but the goal of binning is to produce a computable metric precisely to evaluate how well Eq. 1 is satisfied, mainly because $p(x|c)$ for $c \in \mathbb{R}$  is not well defined. It’s a way to estimate the loss of eq. 3.
> > - Then, you produce an “empirical” version of the loss in eq. 5, but with one term which is an expectation, the conditional accuracy. So, it's not a true empirical criterion.
> > - Then, you approximate this loss by replacing the conditional accuracy with an estimate reduced to a sample, leading to Eq. 6. This step assumes that $1/n \sum_i|c_i - 1/ni \sum_j \{y_{ij} == \hat{y}_{ij}\}| =  1/n \sum_i 1/ni \sum_j |c_i  - \{y_{ij} == \hat{y}_{ij}\}|$ (the summation used for estimating the conditional accuracy is switched before the norm), which is mathematically wrong.
> > - You end up with a sample-based criterion, but which is, for me, precisely the Brier score - this is illustrated by the good performance of your approach on the Brier score.
> >
> > I still struggle to understand the rationale behind your proposed loss: it starts with an ECE-inspired loss and concludes with the Brier score.
> >
> >
> > Q2. OK.
> >
> > Q3. Your approach aims to compute a corrected value of a single scalar value the max softmax score for any new data. I don't understand why you need to express this correction as a modified softmax temperature since the probabilities of the other classes are not involved in the calibration objective.
> >
> > Q4. I still need to see why your empirical loss solves the calibration problem (see my analysis on Q1).
> >
> > Q5. OK
> >
> > Q6. Your answer doesn't address the right issue. Your approach does not calibrate the whole multi-class calibration vector. It only targets the max softmax score, and because you are using a temperature scaling modification, it modifies the values of all scores. However it does not imply that the resulting scores for the other classes will be calibrated.
> >
> > Overall, I have mixed feelings about the article. I found the test-time calibration approach interesting, and giving good results. However, I found the justification of the strategy not well founded, making the results not clearly explainable.

---

> ### Author Response · Authors · 2024-11-25
>
> ### **Q1. Formal Development**
>
> **\\( p(x \mid c) \\) is not well-defined.**
> Thank you for your comment. \\( p(x \mid c) \\) represents the probability distribution of \\( x \\), where the confidence of \\( x \\) is equal to \\( c \\).
>
> **Equation 5 is not a true empirical criterion.**
> To obtain the empirical calibration loss, we gradually approximate Equation (3) through empirical estimation. Equation (5) is an intermediate step in the discretization process. We will revise this part to make it more clear.
>
> **In Equation 6, the summation used for estimating the conditional accuracy is switched before the norm.**
> Thank you for pointing out the issue in the transition from Eq. (5) to Eq. (6). We acknowledge that the transition from Eq. (5) to Eq. (6) is not a strict mathematical equivalence but rather an empirical approximation. We have revised it to make the explanation more rigorous.
>
> On the other hand, whether the summation is inside or outside the norm, both are approximations of the calibration loss, and the development from Eq. (6) to Eq. (7) remains the same. Therefore, the loss function we use in Eq. (7) is not affected by this adjustment.
>
> **End up with a sample-based criterion, but which is precisely the Brier score.**
> We respectfully disagree with this statement. The Brier score is the discrepancy between the predicted softmax confidence **vector** and the one-hot ground truth **vector**. In contrast, our CA loss computes the discrepancy between the confidence value and the correctness (0 or 1), which is fundamentally different in its definition and purpose.
>
>
>
>
> ### **Q3. Why do you need to express this correction as a modified softmax temperature since the probabilities of the other classes are not involved in the calibration objective?**
>
> I'd like to clarify why expressing the correction as a modified softmax temperature is essential, even though our calibration objective focuses on the maximum softmax score.
>
> **1. The softmax temperature influences the maximum score directly**
>
> The softmax function computes probabilities based on the relative differences between the logits of all classes. By adjusting the temperature parameter, we scale these differences:
>
> - **Higher Temperature:** Softens the probability distribution, making the probabilities more uniform and reducing the maximum softmax score.
> - **Lower Temperature:** Sharpens the distribution, increasing the maximum softmax score.
>
> This scaling directly affects the maximum softmax score, allowing us to calibrate it toward 1 when the prediction is correct and toward 0 when it's incorrect.
>
> **2. Maintaining a valid probability distribution**
>
> Adjusting the temperature modifies all class probabilities while ensuring they still sum to 1. This is crucial because:
>
> - **Probabilistic Integrity:** We maintain a valid probability distribution, which is important for interpretability and further probabilistic reasoning.
> - **Consistency Across Predictions:** It ensures that the calibration does not produce anomalous probability distributions that could negatively affect downstream tasks or metrics.
>
> **3. Smooth and principled adjustment mechanism**
>
> Using the temperature parameter allows for a smooth and continuous adjustment of the softmax outputs:
>
> - **Avoiding Abrupt Changes:** Directly altering the maximum score without considering other classes could lead to abrupt or unprincipled changes in the probability distribution.
> - **Theoretical Foundation:** Temperature scaling is a well-established method in calibration literature (e.g., Guo et al., 2017), providing a theoretically sound approach to adjust model confidence.
>
> I hope this explanation clarifies the necessity of expressing the correction as a modified softmax temperature.
>
> - Guo, C., Pleiss, G., Sun, Y., & Weinberger, K. Q. (2017). *On Calibration of Modern Neural Networks*. Proceedings of the 34th International Conference on Machine Learning, PMLR 70:1321-1330.
>
>
> ### **Q4. Why the empirical loss solves the calibration problem.**
>
> The CA loss in Eq. (7) is an approximation of the **integral** (Eq. (3)) of the density function (Eq. (2)), where Eq. (2) is based on the definition of model calibration (Eq. (1)). Therefore, minimizing the CA loss is a step towards achieving model calibration.

---

> ### Author Response · Authors · 2024-11-26
>
> ### **Q6. The temperature modifies the values of all scores. However it does not imply that the resulting scores for the other classes will be calibrated.**
>
> ### **Part 1**
>
>
> Thank you for your valuable feedback and for bringing up an important point regarding the calibration of the entire multi-class probability vector. We provide a detailed explanation and clarify how our method impacts the calibration of all class probabilities, not just the maximum softmax score.
>
> **Gradient computations:**
>
> Our method employs sample-wise post-hoc temperature scaling, where we adjust the temperature parameter \\( T \\) for each sample to calibrate the model's confidence. The key idea is to make the maximum softmax confidence \\( c \\) close to 1 if the prediction is correct and close to 0 if the prediction is incorrect. Our loss function is defined as:
>
> \\[
> L = (c - \text{correctness})^2,
> \\]
>
> where \\( c = \max_i p_i \\) is the maximum softmax probability, and \\(\text{correctness}\\) is 1 if the prediction is correct and 0 otherwise.
>
> Since the temperature scaling affects all class probabilities due to the nature of the softmax function, optimizing \\( T \\) based on this loss function indirectly calibrates the probabilities of all classes. To demonstrate this, we provide a detailed gradient analysis.
>
> The temperature-scaled softmax probabilities are given by:
>
> \\[
> p_i = \frac{\exp\left(\frac{z_i}{T}\right)}{\sum_{j} \exp\left(\frac{z_j}{T}\right)},
> \\]
>
> where \\( z_i \\) are the logits for each class, and \\( T > 0 \\) is the temperature parameter.
>
> The gradient of the loss function \\( L \\) with respect to \\( T \\) is:
>
> \\[
> \frac{\partial L}{\partial T} = 2(c - \text{correctness}) \cdot \frac{\partial c}{\partial T}.
> \\]
>
> Thus, the key term to compute is \\( \frac{\partial c}{\partial T} \\), which depends on the gradient of the maximum softmax probability with respect to \\( T \\).
>
> Computing \\( \frac{\partial c}{\partial T} \\):
>
> Let \\( k = \arg\max_i p_i \\) be the index of the maximum softmax probability. Then:
>
> \\[
> c = p_k = \frac{\exp\left(\frac{z_k}{T}\right)}{\sum_{j} \exp\left(\frac{z_j}{T}\right)}.
> \\]
>
> The derivative of \\( c \\) with respect to \\( T \\) is:
>
> \\[
> \frac{\partial c}{\partial T} = \frac{\partial p_k}{\partial T} = p_k \left( \frac{\sum_{j} p_j \left( \frac{z_j}{T} \right) }{T} - \frac{z_k}{T^2} \right).
> \\]
>
> Simplifying, we get:
>
> \\[
> \frac{\partial c}{\partial T} = \frac{p_k}{T^2} \left( \sum_{j} p_j z_j - z_k \right).
> \\]
>
> Since \\( \sum_{j} p_j z_j \\) is the expected value of the logits under the probability distribution \\( p_j \\), the term \\( \sum_{j} p_j z_j - z_k \\) represents the difference between the expected logit and the maximum logit.
>
> ### **(See Part 2 for the rest of the content.)**

---

> ### Author Response · Authors · 2024-11-26
>
> ### **Q6 Part2 (continued)**
>
> **Case Analysis:**
>
> 1. Correct predictions (\\( \text{correctness} = 1 \\)):
>
>    - Objective: Increase \\( c \\) towards 1.
>    - Gradient direction: Since \\( c < 1 \\) and \\( \text{correctness} = 1 \\), the term \\( (c - \text{correctness}) \\) is negative.
>    - Sign of \\( \frac{\partial c}{\partial T} \\): The term \\( \sum_{j} p_j z_j - z_k \\) is negative because \\( z_k \\) (the logit of the correct class) is higher than the expected logit \\( \sum_{j} p_j z_j \\). Therefore, \\( \frac{\partial c}{\partial T} \\) is negative.
>    - Overall gradient: The product \\( (c - \text{correctness}) \cdot \frac{\partial c}{\partial T} \\) is positve (negative times negative), resulting in \\( \frac{\partial L}{\partial T} > 0 \\).
>    - Effect on \\( T \\): A positive gradient \\( \frac{\partial L}{\partial T} \\) pushes \\( T \\) to decrease during optimization.
>    - **Impact on all class probabilities:** Decreasing \\( T \\) sharpens the softmax distribution, increasing the confidence \\( p_k \\) in the correct class and **decreasing the probabilities \\( p_i \\) of the other classes**, meaning that the other classes are also calibrated. Therefore, we achieve a calibration effect on other classes similar to what Cross-Entropy provides.
>
> 2. Incorrect predictions (\\( \text{correctness} = 0 \\)):
>
>    - Objective: Decrease \\( c \\) towards 0.
>    - Gradient direction: Since \\( c > 0 \\) and \\( \text{correctness} = 0 \\), the term \\( (c - \text{correctness}) \\) is positive.
>    - Sign of \\( \frac{\partial c}{\partial T} \\): The term \\( \sum_{j} p_j z_j - z_k \\) is negative because \\( z_k \\) (the logit of the incorrectly predicted class) is higher than the expected logit. Thus, \\( \frac{\partial c}{\partial T} \\) is negative.
>    - Overall gradient: The product \\( (c - \text{correctness}) \cdot \frac{\partial c}{\partial T} \\) is negative (positive times negative), resulting in \\( \frac{\partial L}{\partial T} < 0 \\).
>    - Effect on \\( T \\): A negative gradient \\( \frac{\partial L}{\partial T} \\) pushes \\( T \\) to increase during optimization.
>    - **Impact on all class probabilities**:  Increasing \( T \) smoothens the softmax distribution, reducing the confidence \( p_k \) in the incorrectly predicted class. For other classes, the ideal outcome is to increase the confidence of the ground-truth class while decreasing the remaining ones. However, it is important to be clear about the constraints of scaling-based post-hoc calibration: it cannot change the order of the logits, meaning it cannot perfectly calibrate each class as expected by cross-entropy loss. It can only make the softmax vector sharper or smoother. As a result, it is challenging to increase the confidence value of the ground-truth class due to the limitations of the softmax operation. More importantly, cross-entropy may result in an undesired temperature, potentially making the incorrect prediction more confident (as shown in Fig. 2), even though cross-entropy seemingly calibrates across all classes directly. In contrast, our method focuses on reducing the maximum confidence value, which is a practical approach to significantly reduce the calibration error, particularly in incorrectly predicted samples.
>
>
> **Experimental Evidence:**
>
> We evaluated the calibration performance of our method using metrics that consider the entire probability distribution:
>
> - Expected Calibration Error (ECE): A lower ECE means that the predicted probabilities more accurately reflect the true likelihood of each class, not just the maximum.
>
> - Brier Score: The Brier Score penalizes both overconfidence and underconfidence across all classes, so improvements here indicate better calibration of the entire probability vector.
>
> Thank you again for your insightful comments. Feel free to reach out with any additional suggestions or questions.
>
> Sincerely,
>
> The Authors

---

> > ### Comment · Reviewer_bjTY · 2024-11-29
> > **Final comments**
> >
> > Thank you for responding in detail to my concerns. Here are my final remarks:
> >
> > Q1: I understand that there are potential connections between the CA loss and the calibration objective, but you do not convincingly prove that they are approximations of the calibration equation, and in what sense. Regarding my interpretation of the CA loss as a Brier score, I agree that this is not the original formulation. However, I think that your loss can be interpreted as the Brier score of the correctness prediction (the prediction being the binary decision whether the classifier outputs the correct class or not).
> >
> > Q3 & Q6: I agree that the temperature, which is instance dependent, can change the maximum prediction score for the target input and consequently the scores for the other classes. However, you are not using the full probability vector for evaluation (since it is not mentioned, I assume you are using the binary ECE metric, which is standard in the field, rather than the class-wise version), and so using a global temperature is not necessary. However, I agree that using the probability vector in the gradient descent can have a smoothing effect (I can see this from your calculation of $\partial c / \partial T$). The additional calculations give some insight into the impact of correctness, but they do not show that the CA loss actually solves the calibration problem when the number of samples goes to infinity, for example.
> >
> > Overall, my opinion has not changed much after the discussion: the approach has interesting elements but the formal justification is weak. I keep my rating.

---

> > > ### Author Response · Authors · 2024-11-29
> > >
> > > ## **Q1. (1) Do not convincingly prove that the CA loss are approximations of the calibration equation, and in what sense.**
> > >
> > > We appreciate your observation that CA Loss is related to the calibration objective, but you found the connection not sufficiently convincing. To clarify, we would like to restate the derivation and explicitly highlight how CA Loss is constructed as an empirical approximation of the theoretical calibration error.
> > >
> > > The original calibration objective (Equation 3) is defined as the expected discrepancy between the predicted confidence \\(\hat{c}\\) and the true conditional accuracy \\(\mathbb{E}^{\text{acc}}\_{\hat{c}}\\):
> > > \\[
> > > E_f = \int l_f(\hat{c}) dp(\hat{c}),
> > > \\]
> > > where \\( l\_f(\hat{c}) = \\|\hat{c} - \\mathbb{E}^{\text{acc}}\_{\hat{c}} \\|\\). In practice, the true conditional distribution \\(p(x \mid \hat{c})\\) is inaccessible. To approximate this, we replaced \\(p(\hat{c})\\) with a Dirac delta function-based empirical distribution:
> > > \\[
> > > dp(\hat{c}) = \frac{1}{n} \sum_{i=1}^n \delta_{\hat{c}\_i}(\hat{c}),
> > > \\]
> > > where \\(\delta_{\hat{c}\_i}(\hat{c})\\) centers the probability mass at each predicted confidence \\(\hat{c}\_i\\). Substituting this into Equation (3), we obtain the empirical calibration error:
> > > \\[
> > > E_f^{\text{emp}} = \frac{1}{n} \sum_{i=1}^n \\|\hat{c}\_i - \mathbb{E}^{\text{acc}}\_{\hat{c}\_i}\\|.
> > > \\]
> > >
> > > The next challenge is the inaccessibility of \\(\mathbb{E}^{\text{acc}}\_{\hat{c}\_i}\\), which represents the true conditional accuracy. To approximate this, we discretize the integral over \\(p(x \mid \hat{c}\_i)\\) as a finite sample average:
> > > \\[
> > > \mathbb{E}^{\text{acc}}\_{\hat{c}\_i} \approx \frac{1}{m} \sum_{j=1}^m \mathbb{I}\\{y\_{x\_{ij}} = \hat{y}\_{x\_{ij}}\\}.
> > > \\]
> > > In most practical datasets, \\(m = 1\\), as there is typically only one sample per predicted confidence level. Thus, \\(\mathbb{E}^{\text{acc}}\_{\hat{c}\_i}\\) is approximated by \\(\mathbb{I}\\{y\_{x\_i} = \hat{y}\_{x\_i}\\}\\). Substituting this into \\(E\_f^{\text{emp}}\\), we derive the CA Loss:
> > > \\[
> > > E\_f^{\text{emp}} = \frac{1}{n} \sum_{i=1}^n \\|\hat{c}\_i - \mathbb{I}\\{y\_{x\_i} = \hat{y}_{x\_i}\\}\\|.
> > > \\]
> > >
> > > **In what sense CA Loss approximates the calibration objective:**
> > > CA Loss approximates the theoretical calibration error in an empirical sense:
> > > - The replacement of the true distribution \\(p(x \mid \hat{c})\\) by its empirical counterpart introduces finite-sample variability.
> > > - As the number of samples increases (\\(n \to \infty\\)), CA Loss converges to the theoretical calibration error under standard assumptions of consistency and representativeness of the dataset.
> > >
> > > To further clarify this, we propose to expand the manuscript with an additional discussion on this approximation, including the assumptions and limitations of finite-sample estimation.
> > >
> > >
> > >
> > >  ## **Q1 (2). Relation to Brier Score**
> > >
> > > Thank you for your observation.
> > >
> > > The CA loss and Brier Score differ fundamentally in definition:
> > >
> > > - The Brier Score measures the squared error between predicted probabilities and ground-truth labels, considering the entire probability distribution.
> > > - CA Loss evaluates the error between the predicted confidence of the top-1 class and the correctness indicator, directly targeting confidence-accuracy alignment.
> > >
> > > **When to be numerical equivalent:**
> > >
> > > CA Loss and the Brier Score are numerically equivalent **only in binary classification**, where the predicted confidence is the same as the probability assigned to the positive class. For multi-class classification, the Brier Score considers all class probabilities, while CA Loss focuses only on the top-1 confidence.

---

> ### Author Response · Authors · 2024-11-29
>
> ## **Q3 & Q6 (1) Authors use the binary ECE metric, which is standard in the field, rather than the class-wise version, so using a global temperature is not necessary.**.
>
> (1) The classical post-hoc model calibration method, **Temperature Scaling** [1], which is widely acknowledged in the literature (with 6000+ citations), uses a temperature to "globally" scale the entire logits vector (across all classes). The evaluation metric used in their work is also the standard **Expected Calibration Error (ECE)**, which is the same metric that we use. Therefore, we do not believe that our evaluation setup is problematic and we also do not think using the temperature to scale all classes is problematic.
>
> (2) Regarding the class-wise version of the calibration error, we understand that the reviewer might be referring to **Static Calibration Error (SCE)**, which evaluates the calibration error for each class individually. While this metric is not commonly used in existing post-hoc model calibration literature, we have still included its results in our discussion with Reviewer LQKt. Please refer to our discussion with Reviewer LQKt for details on the evaluation results using the class-wise calibration error.
>
> [1] Guo, C., Pleiss, G., Sun, Y. and Weinberger, K.Q., 2017, July. On calibration of modern neural networks. In International conference on machine learning (pp. 1321-1330). PMLR.
>
>
> ## **Q3 & Q6 (2) Calculations give some insight but not show that the CA loss solves the calibration problem when the number of samples goes to infinity.**
>
> We thank you for indicating that our previous discussion provided some insight. Regarding this question, as the sample size approaches infinity, we can access the distribution of \\(p(x \mid \hat{c})\\),  allowing us to directly use Equation (2) to compute the theoretical calibration error.
>
> Thank you again for your comments.
>
> Sincerely,
>
> The Authors

---

### Official Review · Reviewer_pvJo · 2024-11-04

**Soundness:** 3
**Presentation:** 3
**Contribution:** 2
**Rating:** 5
**Confidence:** 4

**Summary:**

The paper introduces two innovative methods to address calibration errors in deep learning predictions: a correctness-aware loss function and a sample transformation technique. The correctness-aware loss function aims to directly minimize calibration error, effectively improving the calibration of misclassified samples by narrowing discrepancies across all classes. Additionally, to boost cross-domain performance, an augmentation-based transformation is applied to calibration samples, enhancing robustness across varied domains. Both methods are implemented in a post-hoc calibration framework, and the proposed algorithm demonstrates state-of-the-art performance, particularly in cross-domain settings.

**Strengths:**

This paper presents a range of validation scenarios to assess the effectiveness of the proposed framework. In numerous cases, the framework achieves state-of-the-art performance, validating the impact of its two novel schemes. The experimental setup and comparisons are thoughtfully designed, with detailed descriptions that enhance clarity and reproducibility. Mathematical derivations are presented comprehensively, and the overall narrative is organized in a way that makes the framework easy to follow and understand, emphasizing key components effectively.

**Weaknesses:**

The paper has several strengths, yet I have some specific concerns that warrant attention:

1. Definition of "Narrow Misclassification":
   The term "narrow misclassification" appears in the abstract, and the correctness-aware (CA) loss is presented as targeting this condition by adjusting predictions across different classes rather than solely reducing confidence in the incorrect class. However, a clear definition of "narrow misclassification" is missing, and it’s challenging to discern how it differs from absolutely wrong samples even after reviewing the derivations. Clear definitions and empirical analysis based on outcomes would help clarify this distinction.

2. Limitations from Augmentation Types Used:
   The transformation component uses augmentations, but it lacks an analysis of how different types of augmentation affect performance across domains. Depending on the augmentation type, the efficacy in cross-domain scenarios may vary. Experimental validation or analysis is needed to determine the diversity of augmentation types required or which specific augmentations are essential.

3. Similarity with Temperature Scaling:
   If the framework were designed with temperature scaling, where the temperature parameter is shared across all classes, it could similarly distribute confidence across classes rather than reducing only the incorrect class's confidence. This raises questions about the uniqueness of the proposed algorithm’s approach in addressing "narrow misclassification."

4. Derivation for the CA Loss Function:
   The derivation of the CA loss function appears to be unnecessarily complex. Initially, the paper emphasizes the use of continuous calibration error rather than Expected Calibration Error (ECE), suggesting a different approach. However, the final derivation seems equivalent to ECE-based loss, assuming discrete samples and small sample sizes, which undermines the rationale for a continuous assumption. Clarification is needed on why continuous assumptions were initially made if the final derivation closely resembles an ECE-based approach.

5. Bounds of the CA Loss:
   Bounds for the CA loss are derived based on assumptions that the sample sizes and accuracy across classes are similar. However, the significance of these bounds remains unclear, as they appear merely descriptive of the assumed conditions. Additional insights or generalized bounds demonstrating reduced CE loss could improve understanding.

6. Unclear Derivation in Equation 15:
   The derivation in Equation 15 is ambiguous due to an unexplained arrow, which might imply a limit. Clarification on which parameter converges to produce this outcome is necessary to improve the transparency of this mathematical derivation.

7. Parameter \theta in Equation 19:
   It is unclear if \theta in Equation 19 exclusively refers to the fully connected layers added for post-hoc calibration. This specification is important for clarity.

8. Synergy between CA Loss and Transformation Component:
   The CA loss reduces ECE, while the transformation improves cross-domain robustness. However, the synergy between these components is unclear, as seen in experimental results: applying CA loss significantly reduces ECE, while the transformation tends to increase ECE, showing a trade-off rather than synergy. Clarification is needed on why these mechanisms must be combined rather than sequentially applied as separate approaches.

9. Baseline (CE Only + PTS) Already Achieving State-of-the-Art Performance:
   In the result tables, the baseline (CE Only + PTS) already achieves state-of-the-art ECE and accuracy in multiple scenarios. While adding CA and transformation components improves performance further, it seems that these improvements are achieved largely because of the baseline's strong performance. To mitigate this concern, I recommend testing the proposed algorithm on alternative baselines.

10. Minor Points:
    - The text in figures is too small, making them hard to read.
    - Typo: Line 136, “samples'.” should be “samples.'”

These concerns, if addressed, could enhance the clarity and impact of the proposed framework.

**Questions:**

Why does the paper initially emphasize using a continuous calibration error instead of the Expected Calibration Error (ECE)?

What is the intended synergy between the CA loss and the transformation component, given their distinct purposes of reducing ECE and enhancing cross-domain robustness?

Could the proposed algorithm’s effectiveness be validated further by testing it on alternative baselines?

---

> ### Author Response · Authors · 2024-11-22
>
> ### **Q1. A clear definition of "narrow misclassification" is missing. How it differs from absolutely wrong samples.**
>
> Thank you for raising your confusion.
> We have defined "narrowly wrong sample/prediction" (which has the same meaning as "narrow misclassification") in lines 348 and 377. Additionally, we provided the definition of "absolutely wrong samples" in line 342.
>
> The examples in Fig. 3 also offer a comprehensive explanation of why "narrowly wrong predictions" can have a negative impact during learning when using traditional Maximum Likelihood Estimation methods (e.g., cross-entropy loss).
>
> ### **Q2. Lack of analysis of how different types of augmentation affect performance across domains.**
>
> 1. Our method does not impose constraints on the selection of augmentations, as illustrated in Figure 2. The optimization of augmentation combinations is beyond the scope of this paper.
>
> 2. Considering the vast number of potential datasets in the real world, determining the "best" augmentation combination would be impractical and of limited value. Instead, our experiments are designed to demonstrate that using multiple augmentations consistently yields better results than using a single augmentation (Figure 4), particularly in terms of improving calibration with our CA loss. However, augmentations beyond three do not necessarily lead to better performance.
>
> ### **Q3. The uniqueness of the proposed algorithm’s approach in addressing "narrow misclassification."**
>
> The reviewer may not have fully appreciated the challenge of narrowly incorrect predictions in Maximum Likelihood Estimation (MLE) or the detailed workings of our proposed method.
>
> - MLE always aims to maximize the confidence in the ground truth class. For a "narrowly wrong prediction" (also known as "narrow misclassification"), if we use a temperature to adjust the logits of all classes to achieve higher confidence in the ground truth class, the converged temperature may be less than 1 (as shown in Fig. 3). This can result in the prediction confidence (i.e., the confidence in the wrongly predicted class) for this narrowly wrong sample becoming higher after calibration.
>
> - In contrast, for any wrong prediction—whether it is a narrowly wrong prediction or an absolutely wrong prediction—our method ensures that the temperature converges to a value greater than 1 on expectation. By directly adjusting the logit value of the ground truth class, our approach aims to reduce the confidence of wrong predictions, aligning with the objective derived from the definition of calibration.
>
> ### **Q4. Why continuous assumptions were initially made if the final derivation closely resembles an ECE-based approach.**
>
> The conventional bin-based ECE is widely considered an important metric for evaluating calibration, but technically it cannot be used as a training loss (non-differentiability). Therefore, existing methods usually use conventional cross-entropy loss (e.g., TS (Guo et al., 2017), PTS (Tomani et al., 2022), Adaptive TS (Joy et al., 2023)), which we have discussed has issues with narrowly wrong predictions.
>
> - Our method begins by using the formal definition of model calibration and derives the expected calibration error in a distributional sense. From there, we gradually discretize the calibration error to formulate our Equation 7.
>
> - While Equation 7 serves as an approximation of the expectation of calibration error, it can be directly optimized for calibrator training. Although it appears similar to a continuous ECE formulation, its derivation is fundamentally different, originating from a distributional perspective rather than the traditional ECE evaluation metric.
>
> ### **Q5. The significance of the bounds of CA loss remains unclear**
>
> The significance of the bounds for the CA loss lies in understanding its behavior during training:
>
> - **Lower bound**: Demonstrates whether the loss function can converge during the optimization process, ensuring stability and effectiveness.
>
> - **Upper bound**: Indicates the maximum possible value of the loss, useful for diagnosing training stability and ensuring expected behavior.
>
> Establishing both bounds validates the reliability and applicability of the proposed loss function.
>
>
> ### **Q6. The derivation in Equation 15 is ambiguous due to an unexplained arrow.**
>
> The expression $\frac{1-\rho}{C}$ denotes the lower bound of $\mathbb{E}_f^\text{emp}$, as shown in Eq. (11). The arrow ($\rightarrow$) indicates the optimization of $\mathbb{E}_f^\text{emp}$ towards this lower bound, $\frac{1-\rho}{C}$, which is equivalent to minimizing $\mathbb{E}^\text{diff}$ towards the lower bound $\frac{\Big(\frac{1}{C} - 1\Big)\rho}{1-\rho}$. This process involves pushing the average maximum softmax scores of incorrectly classified samples away from those of correctly classified samples, thereby reducing the overlap of confidence values between correct and incorrect predictions (Lines 250-254 of our main paper).

---

> ### Author Response · Authors · 2024-11-22
>
> ### **Q7. Parameter $\theta$ in Equation 19**
>
> In Eq. 19, $\theta$ represents the calibrator weights, as noted in Line 298 of the main paper.
>
> ### **Q8. What is the synergy between CA Loss and Transformation Component? Applying CA loss reduces ECE, while the transformation tends to increase ECE, showing a trade-off rather than synergy.**
>
> **Synergy between CA Loss and transformation component:**
> CA Loss helps the calibrator learn to adjust the confidence of both correct and incorrect predictions. This requires the post-hoc calibrator to be aware of the correctness of each prediction. Existing work (Deng et al., 2022) has shown that consistency in model predictions for transformed images effectively indicates prediction correctness. CA Loss provides the learning objective, while augmentation techniques make achieving this objective easier.
>
> **Transformation tends to increase ECE, showing a trade-off rather than synergy:**
>
> We believe there is another way to interpret this.
>
> In Tables 1, 2, and 3, combining CA Loss with confidence values from transformations results in lower ECE than using CA Loss alone in 8 out of 10 datasets, demonstrating that the two components work synergistically rather than exhibiting a trade-off.
>
> ### **Q9. Results on alternative baselines**
>
> Our transformation-based calibrator can unlock the potential of CA loss. Meanwhile, our CA loss also provides positive improvements to other calibrators, as demonstrated in Table 1, 2, and 3, where it enhances PTS.
>
> We also replace the cross-entropy (CE) loss with correctness-aware (CA) loss in the temperature scaling (TS) method. Below, we present our experimantal results on ImageNet-A, ImageNet-R, ImageNet-S and ObjectNet.
>
> ImageNet-A:
>
> | **Method**     | **ECE $\downarrow$** | **BS $\downarrow$** | **KS $\downarrow$** | **AUC $\uparrow$** |
> |----------------|-----------|----------|----------|-----------|
> | **Uncal**      | 39.44     | 32.90    | 43.47    | 61.87     |
> | **TS + CE**    | 29.24     | 23.24    | 32.50    | 62.86     |
> | **TS + CA**    | **24.93** | **19.22**| **27.76**| **63.16** |
>
>
> ImageNet-R:
>
> | **Method**     | **ECE $\downarrow$** | **BS $\downarrow$** | **KS $\downarrow$** | **AUC $\uparrow$** |
> |----------------|-----------|----------|----------|-----------|
> | **Uncal**      | 13.97     | 16.89    | 19.90    | 88.06     |
> | **TS + CE**    | 6.28      | 13.80    | 14.51    | 88.27     |
> | **TS + CA**    | 6.50      | 14.10    | **13.16**| 88.21     |
>
>
> ImageNet-S:
>
> | **Method**     | **ECE $\downarrow$** | **BS $\downarrow$** | **KS $\downarrow$** | **AUC $\uparrow$** |
> |----------------|-----------|----------|----------|-----------|
> | **Uncal**      | 20.92     | 21.67    | 29.01    | 82.57     |
> | **TS + CE**    | 8.92      | 16.11    | 19.18    | 83.22     |
> | **TS + CA**    | **6.22**  | **15.45**| **14.84**| 83.09     |
>
>
> ObjectNet:
>
> | **Method**     | **ECE $\downarrow$** | **BS $\downarrow$** | **KS $\downarrow$** | **AUC $\uparrow$** |
> |----------------|-----------|----------|----------|-----------|
> | **Uncal**      | 31.21     | 25.20    | 36.48    | 78.05     |
> | **TS + CE**    | 19.70     | 17.97    | 26.92    | 78.35     |
> | **TS + CA**    | **12.83** | **14.90**| **21.59**| 78.34 |
>
> Results show that our method also provides improvements. However, the results here are not as significant as those achieved with CA loss combined with our calibrator (with image transformations). This is because the consistency across image transformations provides informative features that help achieve the learning objective of CA loss.
>
>
> ### **Q10. Minor points**
>
> Thanks for your suggestion. We have addressed them in our revised paper.
>
>
> ### **Q11. Why the paper emphasizes using a continuous calibration error instead of the Expected Calibration Error (ECE)?**
>
> Kindly refer to our response in **Q4**.
>
> ### **Q12. What is the intended synergy between the CA loss and the transformation component?**
>
> Kindly refer to our response in **Q8**.
>
>
> ### **Q13. Results on alternative baselines**
>
> Kindly refer to our response in **Q9**.

---

> ### Comment · Reviewer_pvJo · 2024-11-28
>
> The meanings of the previously confused terms have been clarified after the rebuttal. I hope the authors will revise the paper to better explain the definitions provided in the comments. Specifically, I expect the additional measurements to be included in the appendix or elsewhere. The additional experiments impressively demonstrate the generalizability of the proposed algorithm. Believing that these points will be addressed, I have decided to increase my score.

---

> > ### Author Response · Authors · 2024-11-28
> >
> > Dear Esteemed Reviewer pvJo,
> >
> > We sincerely thank you for your feedback and questions, which have been invaluable in improving our work.
> >
> > We are also grateful to know that the concerns you raised have been addressed, and we appreciate your decision to raise the rating of our submission.
> >
> > Rest assured, we will carefully revise the paper, incorporating all your suggestions to enhance its clarity, quality, and overall strength.
> >
> > Thank you once again for your time and efforts in reviewing our work.
> >
> > Sincerely,
> > The Authors

---

### Meta-Review · Area_Chair_1D6c · 2024-12-23

**Metareview:**

This paper received mixed reviews. The reviewers recognized the novel calibration loss and its sound motivation, extensive experiments, and competitive performance the paper achieved. They also raised various concerns, some of which are crucial; to name a few: weak theoretical justification (bjTY), presentation issues (e.g., notations and terms not defined properly, unnecessarily complex derivation of the loss, issues on global logic) (pvJo, bjTY), the use of input transformations (e.g., dependency on a set of transformations, lack of justification and theoretical foundation) (pvJo, bjTY, iPzo), potential trade-off between the two proposed components (pvJo), the use of an overly strong baseline (pvJo), and marginal difference from previous work (LQKt).

The authors' rebuttal and subsequent responses in the discussion period address some of these concerns but failed to fully assuage all of them: two reviewers (pvJo, bjTY) still pointed out the weak theoretical justification and presentation issues. As a result, the two reviewers voted down after the discussion period. The AC found that these remaining concerns are not trivial and should be addressed appropriately before publication. In particular, Reviewer bjTY noted that the theoretical justification is weak since the derivation of the loss relies on an assumption that does not hold in general; this is a critical limitation since the derivation is central to the theoretical arguments made by the authors and, according to them, one of the main contributions of the paper. Moreover, the AC found that many critical concerns of the reviewers have not been well addressed in the rebuttal and revision. For example, although a couple of reviewers pointed out presentation issues ranging from unclear definitions of notations and terms to weak global logic, the manuscript has not been revised to address these concerns. Also, although some concerns like impact of input transformations and trade-off between the CA loss and the transformations can be addressed by simple and straightforward experiments, the authors did not directly address the concern, but reiterated their arguments using some of the reported results.

Putting these together, the AC considers that the remaining concerns outweigh the positive comments and the rebuttal, and thus regrets to recommend rejection. The authors are encouraged to revise the paper with the comments by the reviewers, and submit to an upcoming conference.

**Additional Comments On Reviewer Discussion:**

- **Weak theoretical justification (bjTY)**: *This is the most serious concern and the main objection of the AC.* The reviewer found that the mathematical derivation of the proposed loss, which is one of the main contributions of the paper according to the authors, relies on an assumption that does not hold in general. The authors failed to assuage this concern.
- **Presentation issues (pvJo, bjTY)**: The two reviewers, in particular pvJo, raised several critical concerns with the weak quality of presentation, but the authors failed to assuage these concerns and did not reflect the comments in the revision. Examples include unclear definitions of some math notations and terms (pvJo, bjTY), unclear logic (bjTY), unnecessarily complex derivation of the loss (pvJo), and unclear motivation of deriving the bounds of the CA loss (pvJo). The AC found that these comments are valid and can substantially improve the quality of writing if properly addressed in the revision. *At the same time, the AC sees that the volume of required revision will exceed what is typically expected from a camera-ready revision.*
	- Unclear definitions of some math notations and terms (pvJo, bjTY): The abstract should be revised appropriately to explain even briefly "narrowed misclassification", which has not been widely used in the literature.
	- Hard-to-follow logic (bjTY): The response looks making sense but the paper was not revised to address this issue; actually, the paper has to be largely rewritten and thus the volume of revision will clearly exceed what we usually expect from a camera-ready revision.
	- Unnecessarily complex derivation of the loss (pvJo): The rationale was described in the rebuttal but its not crystal clear. Also, it was not added to the revision.
	- Unclear motivation of deriving the bounds of the CA loss (pvJo): The authors' response does not sound convincing. The authors stated that the bounds validates the reliability and applicability of the proposed loss function, but the paper does not have any analysis or experiments regarding/using such bounds. Also, the roles of the upper and lower bounds of the loss have not been described in the revision.
- **Concerns with the use of transformations (pvJo, bjTY, iPzo)**: The reviewers wondered how much sensitive the proposed method is to the transformation types, and asks theoretical foundation of the use of transformations. Their comments are indeed valid, and they did not blame the authors but just wanted to understand the behavior of the model more thoroughly. Moreover, some of the concerns can be straightforwardly addressed by an empirical investigation. However, the authors did not directly address the concerns and did not conduct even a simple experiment.
- **Trade-off between the two proposed components, the CA loss and input transformations (pvJo)**: The authors did not directly address the reviewer's concern, but stated what they want to say (which is a bit ambiguous) using some of the reported results. The AC believes the authors should present even a simple empirical evidences demonstrating that the two components do not contradict each other.
- Remaining concerns include the use of an overly strong baseline (pvJo), missing references (LQKt), marginal difference from previous work (LQKt),  and no reliability diagram reported (LQKt). These concerns have been successfully resolved by the rebuttal.

---

### Decision · Program_Chairs · 2025-01-22

Reject